# PREDICTIVE INFERENCE WITH FEATURE CONFORMAL PREDICTION

**Jiaye Teng**[1,3,4,*], **Chuan Wen**[1,3,4,*], **Dinghuai Zhang**[2,*],
**Yoshua Bengio**[2], **Yang Gao**[1,3,4], **Yang Yuan**[1,3,4,†]
[1]Institute for Interdisciplinary Information Sciences, Tsinghua University
[2]Mila - Quebec AI Institute
[3]Shanghai Artificial Intelligence Laboratory
[4]Shanghai Qi Zhi Institute
`{tjy20,cwen20}@mails.tsinghua.edu.cn, dinghuai.zhang@mila.quebec`

## ABSTRACT

Conformal prediction is a distribution-free technique for establishing valid prediction intervals. Although conventionally people conduct conformal prediction in the output space, this is not the only possibility. In this paper, we propose feature conformal prediction, which extends the scope of conformal prediction to semantic feature spaces by leveraging the inductive bias of deep representation learning. From a theoretical perspective, we demonstrate that feature conformal prediction provably outperforms regular conformal prediction under mild assumptions. Our approach could be combined with not only vanilla conformal prediction, but also other adaptive conformal prediction methods. Apart from experiments on existing predictive inference benchmarks, we also demonstrate the state-of-the-art performance of the proposed methods on *large-scale* tasks such as ImageNet classification and Cityscapes image segmentation. The code is available in `https://github.com/AlvinWen428/FeatureCP`.

## 1 INTRODUCTION

Although machine learning models work well in numerous fields (Silver et al., 2017; Devlin et al., 2019; Brown et al., 2020), they usually suffer from over-confidence issues, yielding unsatisfactory uncertainty estimates (Guo et al., 2017a; Chen et al., 2021; Gawlikowski et al., 2021). To tackle the uncertainty issues, people have developed a multitude of uncertainty quantification techniques, including calibration (Guo et al., 2017b; Minderer et al., 2021), Bayesian neural networks (Smith, 2014; Blundell et al., 2015), and many others (Sullivan, 2015).

Among different uncertainty quantification techniques, *conformal prediction* (CP) stands out due to its simplicity and low computational cost properties (Vovk et al., 2005; Shafer & Vovk, 2008; Angelopoulos & Bates, 2021). Intuitively, conformal prediction first splits the dataset into a training fold and a calibration fold, then trains a machine learning model on the training fold, and finally constructs the confidence band via a non-conformity score on the calibration fold. Notably, the confidence band obtained by conformal prediction is *guaranteed* due to the exchangeability assumption in the data. With such a guarantee, conformal prediction has been shown to perform promisingly on numerous realistic applications (Lei & Candès, 2021b; Angelopoulos et al., 2022).

Despite its remarkable effectiveness, vanilla conformal prediction (vanilla CP) is only deployed in the output space, which is not the only possibility. As an alternative, feature space in deep learning stands out due to its powerful inductive bias of deep representation. Take the image segmentation problem as an example. In such problems, we anticipate a predictive model to be certain in the informative regions (*e.g.*, have clear objects), while uncertain elsewhere. Since different images would possess different object boundary regions, it is inappropriate to return the same uncertainty for different positions, as standard conformal prediction does. Nonetheless, if we instead employ conformal

---

*Equal Contribution.
†Correspond to `yuanyang@mail.tsinghua.edu`.

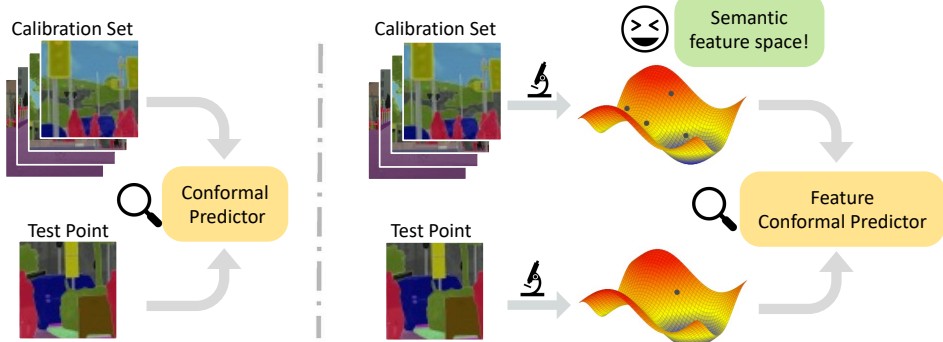

Figure 1: Illustration of vanilla CP (left) vs Feature CP (right). Feature CP operates in the semantic feature space, as opposed to the commonly adopted output space. These methods are described in further detail in Sections 3 and 4.

prediction on the more meaningful feature space, albeit all images have the same uncertainty on this intermediate space, the pixels would exhibit effectively different uncertainty in the output space after a non-trivial non-linear transformation (see Figure 3).

In this work, we thus propose the Feature Conformal Prediction (Feature CP) framework, which deploys conformal prediction in the feature space rather than the output space (see Figure 1). However, there are still two issues unsolved for performing Feature CP: (a) commonly used non-conformity scores require a ground truth term, but here the ground truth in feature space is not given; and (b) transferring the confidence band in the feature space to the output space is non-trivial. To solve problem (a), we propose a new non-conformity score based on the notation surrogate feature, which replaces the ground truth term in previous non-conformity scores. As for (b), we propose two methods: *Band Estimation*, which calculates the upper bound of the confidence band, together with *Band Detection*, to determine whether a response locates in the confidence band. More interestingly, feature-level techniques are pretty general and can be deployed into other distribution-free inference algorithms, *e.g.*, conformalized quantile regression (CQR). This shows the great potential application impact of the proposed Feature CP methodology (see the discussion in Appendix B.4).

From a theoretical perspective, we demonstrate that *Feature CP is provably more efficient, in the sense that it yields shorter confidence bands than vanilla CP, given that the feature space meets cubic conditions*. Here the cubic conditions sketch the properties of feature space from three perspectives, including length preserving, expansion, and quantile stability (see Theorem 6). At a colloquial level, the cubic conditions assume the feature space has a smaller distance between individual non-conformity scores and their quantiles, which reduces the cost of the quantile operation. We empirically validate that the feature space in deep learning satisfies the cubic conditions, thus resulting in a better confidence band with a shorter length (See Figure 2) according to our theoretical analysis.

Our contributions can be summarized as follows:

- We propose Feature CP, together with a corresponding non-conformity score and an uncertainty band estimation method. The proposed method no longer treats the trained model as a black box but exploits the semantic feature space information. What's more, our approach could be directly deployed with any pretrained model as a plug-in component, without the need of re-training under specially designed learning criteria.

- Theoretical evidence guarantees that Feature CP is both (a) efficient, where it yields shorter confidence bands, and (b) effective, where the empirical coverage provably exceeds the given confidence level, under reasonable assumptions.

- We conduct extensive experiments under both synthetic and realistic settings (*e.g.*, pixel-level image segmentation) to corroborate the effectiveness of the proposed algorithm. Besides, we demonstrate the universal applicability of our method by deploying feature-level operations to improve other adaptive conformal prediction methods such as CQR.

## 2 RELATED WORK

Conformal prediction is a statistical framework dealing with uncertainty quantification problems (Vovk et al., 2005; Shafer & Vovk, 2008; Nouretdinov et al., 2011; Barber et al., 2020; Angelopoulos & Bates, 2021). The research on conformal prediction can be roughly split into the following branches. The first line of work focuses on relaxing the assumptions about data distribution in conformal prediction, *e.g.*, exchangeability (Tibshirani et al., 2019; Hu & Lei, 2020; Podkopaev & Ramdas, 2021; Barber et al., 2022). The second line aims at improving the efficiency of conformal prediction (Romano et al., 2020b; Sesia & Candès, 2020; Izbicki et al., 2020a; Yang & Kuchibhotla, 2021; Stutz et al., 2021). The third line tries to generalize conformal prediction to different settings, *e.g.*, quantile regression (Romano et al., 2019), $k$-Nearest Neighbors (Papadopoulos et al., 2011), density estimator (Izbicki et al., 2020b), survival analysis (Teng et al., 2021; Candès et al., 2021), or conditional histogram regression (Sesia & Romano, 2021). There are also works combining conformal prediction with other machine learning topics, such as functional data (Lei et al., 2013), treatment effects (Lei & Candès, 2021a), time series analysis (Xu & Xie, 2021), online learning (Gibbs & Candès, 2021), adversarial robustness (Gendler et al., 2022), and many others.

Besides conformal prediction, there are many other uncertainty quantification techniques, including calibration (Guo et al., 2017a; Kuleshov et al., 2018; Nixon et al., 2019) and Bayesian-based techniques (Blundell et al., 2015; Hernández-Lobato & Adams, 2015; Li & Gal, 2017). Different from the above techniques, conformal prediction is appealing due to its simplicity, computationally free, and model-free properties.

Image segmentation is a traditional task in computer vision, which focuses on partitioning images into different semantic segments (Haralick & Shapiro, 1985; Senthilkumaran & Rajesh, 2009; Minaee et al., 2020). A line of researches applies conformal prediction with some threshold output for all pixels (Angelopoulos & Bates, 2021; Bates et al., 2021), or focus on the risk control tasks (Angelopoulos et al., 2021a). Different from previous approaches, our method first achieves meaningful pixel-level conformal prediction results to the best of our knowledge.

## 3 PRELIMINARIES

**Predictive inference.** Let $(X, Y) \sim \mathcal{P}$ denotes a random data pair, *e.g.*, an image and its segmentation map. Given a significance level $\alpha$, we aim to construct a confidence band $\mathcal{C}_{1-\alpha}(X)$, such that

$$\mathbb{P}\left(Y \in \mathcal{C}_{1-\alpha}(X)\right) \geq 1 - \alpha. \tag{1}$$

There is a tradeoff between efficiency and effectiveness, since one can always set $\mathcal{C}_{1-\alpha}(X)$ to be infinitely large to satisfy Equation (1). In practice, we wish the measure of the confidence band (*e.g.*, its length) can be as small as possible, given that the coverage in Equation (1) holds.

**Dataset.** Let $\mathcal{D} = \{(X_i, Y_i)\}_{i \in \mathcal{I}}$ denotes the dataset, where $\mathcal{I}$ denotes the set of data index and $(X_i, Y_i)$ denotes a sample pair following the distribution $\mathcal{P}$. Typically, conformal prediction requires that data in $\mathcal{D}$ satisfies exchangeability (see below) rather than the stronger i.i.d. (independent and identically distributed) condition. We use $|\mathcal{I}|$ to represent the cardinality of a set $\mathcal{I}$. Conformal prediction needs to first randomly split the dataset into a training fold $\mathcal{D}_{\mathrm{tr}} = \{(X_i, Y_i)\}_{i \in \mathcal{I}_{\mathrm{tr}}}$ and a calibration fold $\mathcal{D}_{\mathrm{ca}} = \{(X_i, Y_i)\}_{i \in \mathcal{I}_{\mathrm{ca}}}$, where $\mathcal{I}_{\mathrm{tr}} \cup \mathcal{I}_{\mathrm{ca}} = \mathcal{I}$ and $\mathcal{I}_{\mathrm{tr}} \cap \mathcal{I}_{\mathrm{ca}} = \phi$. We denote the test point as $(X', Y')$, which is also sampled from the distribution $\mathcal{P}$.

**Training process.** During the training process, we train a machine learning model denoted by $\hat{\mu}(\cdot)$ (*e.g.*, neural network) with the training fold $\mathcal{D}_{\mathrm{tr}}$. For the ease of the following discussion, we rewrite the model as $\hat{\mu} = \hat{g} \circ \hat{f}$, where $\hat{f}$ denotes the feature function (*i.e.*, first several layers in neural networks) and $\hat{g}$ denotes the prediction head (*i.e.*, last several layers in neural networks).

**Calibration process.** Different from usual machine learning methods, conformal prediction has an additional calibration process. Specifically, we calculate a *non-conformity score* $V_i = s(X_i, Y_i, \hat{\mu})$ based on the calibration fold $\mathcal{D}_{\mathrm{ca}}$, where $s(\cdot, \cdot, \cdot)$ is a function informally measuring how the model $\hat{\mu}$ fits the ground truth. The simplest form of non-conformity score is $s(X_i, Y_i, \hat{\mu}) = \|Y_i - \hat{\mu}(X_i)\|$. One could adjust the form of the non-conformity score according to different contexts (*e.g.*, Romano et al. (2019); Teng et al. (2021)). Based on the selected non-conformity score, a matching confidence band could be subsequently created.

---

**Algorithm 1** Conformal Prediction

---

**Require:** Desired confidence level $\alpha$, dataset $\mathcal{D} = \{(X_i, Y_i)\}_{i \in \mathcal{I}}$, test point $X'$, non-conformity score function $s(\cdot)$
 1: Randomly split the dataset $\mathcal{D}$ into a training fold $\mathcal{D}_{\text{tr}} \triangleq (X_i, Y_i)_{i \in \mathcal{I}_{\text{tr}}}$ and a calibration fold $\mathcal{D}_{\text{ca}} \triangleq (X_i, Y_i)_{i \in \mathcal{I}_{\text{ca}}}$;
 2: Train a base machine learning model $\hat{\mu}(\cdot)$ with $\mathcal{D}_{\text{tr}}$ to estimate the response $Y_i$;
 3: For each $i \in \mathcal{I}_{\text{ca}}$, calculate its non-conformity score $V_i = s(X_i, Y_i, \hat{\mu})$;
 4: Calculate the $(1 - \alpha)$-th quantile $Q_{1-\alpha}$ of the distribution $\frac{1}{|\mathcal{I}_{\text{ca}}|+1} \sum_{i \in \mathcal{I}_{ca}} \delta_{V_i} + \delta_{\infty}$.
**Ensure:** $\mathcal{C}_{1-\alpha}(X') = \{Y : s(X', Y, \hat{\mu}) \leq Q_{1-\alpha}\}$.

---

We present vanilla CP[1] in Algorithm 1. Moreover, we demonstrate its theoretical guarantee in Proposition 2, based on the following notation of exchangeability in Assumption 1.

**Assumption 1** (exchangeability). *Assume that the calibration data $(X_i, Y_i), i \in \mathcal{I}_{ca}$ and the test point $(X', Y')$ are exchangeable. Formally, define $Z_i, i = 1, \ldots, |\mathcal{I}_{ca} + 1|$, as the above data pair, then $Z_i$ are exchangeable if arbitrary permutation leads to the same distribution, i.e.,*

$$(Z_1, \ldots, Z_{|\mathcal{I}_{ca}|+1}) \stackrel{d}{=} (Z_{\pi(1)}, \ldots, Z_{\pi(|\mathcal{I}_{ca}|+1)}), \tag{2}$$

*with arbitrary permutation $\pi$ over $\{1, \cdots, |\mathcal{I}_{ca} + 1|\}$.*

Note that Assumption 1 is weaker than the i.i.d. assumption. Therefore, it is reasonable to assume the exchangeability condition to hold in practice. Based on the exchangeability assumption, one can show the following theorem, indicating that conformal prediction indeed returns a valid confidence band, which satisfies Equation (1).

**Theorem 2** (theoretical guarantee for conformal prediction (Law, 2006; Lei et al., 2018; Tibshirani et al., 2019)). *Under Assumption 1, the confidence band $\mathcal{C}_{1-\alpha}(X')$ returned by Algorithm 1 satisfies*

$$\mathbb{P}(Y' \in \mathcal{C}_{1-\alpha}(X')) \geq 1 - \alpha.$$

## 4 METHODOLOGY

In this section, we broaden the concept of conformal prediction using feature-level operations. This extends the scope of conformal prediction and makes it more flexible. We analyze the algorithm components and details in Section 4.1 and Section 4.2. The algorithm is finally summarized in Section 4.3. We remark that although in this work we discuss Feature CP under regression regimes for simplicity's sake, one can easily extend the idea to classification problems.

### 4.1 NON-CONFORMITY SCORE

Conformal prediction necessitates a non-conformity score to measure the conformity between prediction and ground truth. Traditional conformal prediction usually uses norm-based non-conformity score due to its simplicity, *i.e.*, $s(X, Y, \mu) = \|Y - \mu(X)\|$, where $Y$ is the provided ground truth target label. Nonetheless, we have no access to the given target features if we want to conduct conformal prediction at the feature level. To this end, we introduce the *surrogate feature* (see Definition 3), which could serve as the role of ground truth $Y$ in Feature CP.

**Definition 3** (Surrogate feature). *Consider a trained neural network $\hat{\mu} = \hat{g} \circ \hat{f}$ where $\circ$ denotes the composition operator. For a sample $(X, Y)$, we define $\hat{v} = \hat{f}(X)$ to be the trained feature. Besides, we define the surrogate feature to be any feature $v$ such that $\hat{g}(v) = Y$.*

In contrast to commonly adopted regression or classification scenarios where the label is unidimensional, the dimensionality of features could be much larger. We thus define a corresponding non-conformity score based on the surrogate feature as follows:

$$s(X, Y, \hat{g} \circ \hat{f}) = \inf_{v \in \{v : \hat{g}(v) = Y\}} \|v - \hat{f}(X)\|. \tag{3}$$

---

[1]We use $\delta_u$ to represent a Dirac Delta function (distribution) at point $u$.

It is usually complicated to calculate the score in Equation 3 due to the infimum operator. Therefore, we design Algorithm 2 to calculate an upper bound of the non-conformity score. Although the exact infimum is hard to achieve in practice, we can apply gradient descent starting from the trained feature $\hat{v}$ to find a surrogate feature $v$ around it. In order to demonstrate the reasonability of this algorithm, we analyze the non-conformity score distribution with realistic data in Appendix B.9.

## 4.2 BAND ESTIMATION AND BAND DETECTION

Utilizing the non-conformity score derived in Section 4.1, one could derive a confidence band in the feature space. In this section, we mainly focus on how to transfer the confidence band in feature space to the output space, *i.e.*, calculating the set

$$\{\hat{g}(v) : \|v - \hat{v}\| \leq Q_{1-\alpha}\}, \tag{4}$$

where $\hat{v}$ is the trained feature, $\hat{g}$ is the prediction head, and $Q_{1-\alpha}$ is derived based on the calibration set (even though slightly different, we refer to step 4 in Algorithm 1 for the notion of $Q_{1-\alpha}$; a formal discussion of it is deferred to Algorithm 3).

Since the prediction head $\hat{g}$ is usually highly nonlinear, the exact confidence band is hard to represent explicitly. Consequently, we provide two approaches: *Band Estimation* which aims at estimating the upper bound of the confidence band, and *Band Detection* which aims at identifying whether a response falls inside the confidence interval. We next crystallize the two methods.

---

**Algorithm 2** Non-conformity Score

**Require:** Data point $(X, Y)$, trained predictor $\hat{g} \circ \hat{f}(\cdot)$, step size $\eta$, number of steps $M$;
1: $u \leftarrow \hat{f}(X)$;
2: $m \leftarrow 0$;
3: **while** $m < M$ **do**
4: $\quad u \leftarrow u - \eta \frac{\partial \|\hat{g}(u) - Y\|^2}{\partial u}$;
5: $\quad m \leftarrow m + 1$;
6: **end while**
**Ensure:** $s(X, Y, \hat{g} \circ \hat{f}) = \|u - \hat{f}(X)\|$.

---

**Band Estimation.** We model the Band Estimation problem as a perturbation analysis one, where we regard $v$ in Equation (4) as a perturbation of the trained feature $\hat{v}$, and analyze the output bounds of prediction head $\hat{g}$. In this work, we apply linear relaxation based perturbation analysis (LiPRA) (Xu et al., 2020) to tackle this problem under deep neural network regimes. LiPRA transforms the certification problem as a linear programming problem, and solves it accordingly. The relaxation would result in a relatively looser interval than the actual band, so this method would give an upper bound estimation of the exact band length.

**Band Detection.** Band Estimation could potentially end up with loose inference results. Typically, we are only interested in determining whether a point $\tilde{Y}$ is in the confidence band $\mathcal{C}(X')$ for a test sample $X'$. To achieve this goal, we first apply Algorithm 2 using data point $(X', \tilde{Y})$, which returns a non-conformity score $\tilde{V}$. We then test whether the score $\tilde{V}$ is smaller than quantile $Q_{1-\alpha}$ on the calibration set (see Equation (4)). If so, we deduce that $\tilde{Y} \in \mathcal{C}(X')$ (or vice versa if not).

## 4.3 FEATURE CONFORMAL PREDICTION

Based on the above discussion, we summarize[2] Feature CP in Algorithm 3. Different from vanilla CP (see Algorithm 1), Feature CP uses a different non-conformity score based on surrogate features, and we need an additional Band Estimation or Band Detection (step 5) to transfer the band from feature space to output space.

We then discuss two intriguing strengths of Feature CP. First, the proposed technique is universal and could improve other advanced adaptive conformal inference techniques utilizing the inductive bias of learned feature space. Specifically, we propose Feature CQR with insights from CQR (Romano et al., 2019), a prominent adaptive conformal prediction method with remarkable performance, to demonstrate the universality of our technique. We relegate related algorithmic details to Section B.4. Second, although methods such as CQR require specialized training criteria (*e.g.*, quantile regression) for the predictive models, Feature CP could be directly applied to any given pretrained model and

---

[2]We here show the Feature CP algorithm with Band Estimation. We defer the practice details of using Band Detection in step 5 of Algorithm 3 to Appendix.

---

**Algorithm 3** Feature Conformal Prediction

---

**Require:** Level $\alpha$, dataset $\mathcal{D} = \{(X_i, Y_i)\}_{i \in \mathcal{I}}$, test point $X'$;
 1: Randomly split the dataset $\mathcal{D}$ into a training fold $\mathcal{D}_{\text{tr}} \triangleq (X_i, Y_i)_{i \in \mathcal{I}_{\text{tr}}}$ together with a calibration fold $\mathcal{D}_{\text{ca}} \triangleq (X_i, Y_i)_{i \in \mathcal{I}_{\text{ca}}}$;
 2: Train a base machine learning model $\hat{g} \circ \hat{f}(\cdot)$ using $\mathcal{D}_{\text{tr}}$ to estimate the response $Y_i$;
 3: For each $i \in \mathcal{I}_{\text{ca}}$, calculate the non-conformity score $V_i$ based on Algorithm 2;
 4: Calculate the $(1 - \alpha)$-th quantile $Q_{1-\alpha}$ of the distribution $\frac{1}{|\mathcal{I}_{\text{ca}}|+1} \sum_{i \in \mathcal{I}_{ca}} \delta_{V_i} + \delta_\infty$;
 5: Apply Band Estimation on test data feature $\hat{f}(X')$ with perturbation $Q_{1-\alpha}$ and prediction head $\hat{g}$, which returns $\mathcal{C}_{1-\alpha}^{\text{fcp}}(X)$;
**Ensure:** $\mathcal{C}_{1-\alpha}^{\text{fcp}}(X)$.

---

could still give meaningful adaptive interval estimates. This trait facilitates the usage of our method with large pretrained models, which is common in modern language and vision tasks.

### 4.4 THEORETICAL GUARANTEE

This section presents theoretical guarantees for Feature CP regarding coverage (effectiveness) and band length (efficiency). We provide an informal statement of the theorem below and defer the complete details to Appendix A.1.

**Theorem 4** (Informal Theorem on the Efficiency of Feature CP). *Under mild assumptions, if the following cubic conditions hold:*

   1. ***Length Preservation.** Feature CP does not cost much loss in feature space.*

   2. ***Expansion.** The Band Estimation operator expands the differences between individual length and their quantiles.*

   3. ***Quantile Stability.** The band length is stable in both the feature space and the output space for a given calibration set.*

*then Feature CP outperforms vanilla CP in terms of average band length.*

The intuition of Theorem 4 is as follows: Firstly, Feature CP and Vanilla CP take quantile operations in different spaces, and the Expansion condition guarantees that the quantile step costs less in Feature CP. However, there may be an efficiency loss when transferring the band from feature space to output space. Fortunately, it is controllable under the Length Preservation condition. The final Quantile Stability condition ensures that the band is generalizable from the calibration fold to the test samples. We provide the detailed theorem in Appendix A.1 and empirically validate the cubic conditions in Appendix B.2.

## 5 EXPERIMENTS

We conduct experiments on synthetic and real-world datasets, mainly to show that Feature CP is (a) effective, *i.e.*, it could return valid confidence bands with empirical coverage larger than $1 - \alpha$; (b) efficient, *i.e.*, it could return shorter confidence bands than vanilla CP.

### 5.1 SETUP

**Datasets.** We consider both synthetic datasets and real-world datasets, including (a) realistic uni-dimensional target datasets: five datasets from UCI machine learning repository (Asuncion, 2007): physicochemical properties of protein tertiary structure (bio), bike sharing (bike), community and crimes (community) and Facebook comment volume variants one and two (facebook 1/2), five datasets from other sources: blog feedback (blog) (Buza, 2014), Tennessee's student teacher achievement ratio (star) (Achilles et al., 2008), and medical expenditure panel survey (meps19–21) (Cohen et al., 2009); (b) synthetic multi-dimensional target dataset: $Y = WX + \epsilon$, where $X \in [0, 1]^{100}, Y \in \mathbb{R}^{10}$,

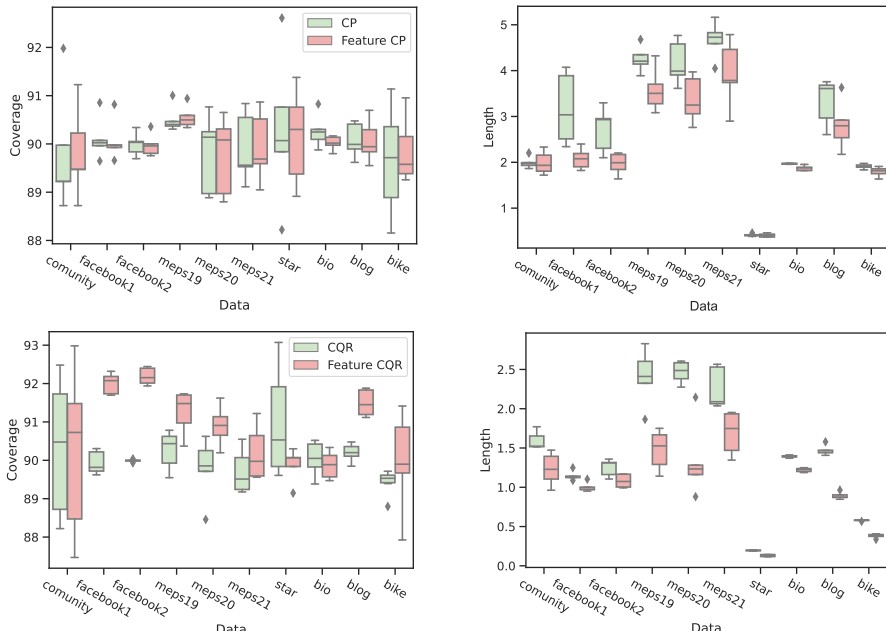

Figure 2: Performance on tasks with one-dimensional target ($\alpha = 0.1$). *Left:* Empirical coverage. *Right:* Confidence interval length, where smaller value is better. The proposed Feature CP and Feature CQR could consistently achieve shorter bands while maintaining a good coverage performance.

Table 1: Performance of both methods on multi-dimensional regression benchmarks ($\alpha = 0.1$), where "w-length" denotes the weighted confidence band length.

| DATASET | SYNTHETIC | | CITYSCAPES | | |
|---|---|---|---|---|---|
| METHOD | COVERAGE | LENGTH | COVERAGE | LENGTH | W-LENGTH |
| BASELINE | $89.91 \pm 1.03$ | $0.401 \pm 0.01$ | $91.41 \pm 0.51$ | $40.15 \pm 0.02$ | $40.15 \pm 0.02$ |
| FEATURE CP | $90.13 \pm 0.59$ | $\mathbf{0.373} \pm 0.05$ | $90.77 \pm 0.91$ | $\mathbf{1.032} \pm 0.01$ | $\mathbf{0.906} \pm 0.01$ |

$\epsilon$ follows the standard Gaussian distribution, and $W$ is a fixed randomly generated matrix; and (c) real-world semantic segmentation dataset: Cityscapes (Cordts et al., 2016), where we transform the original pixel-wise classification problem into a high-dimensional pixel-wise regression problem. We also extend Feature CP to classification problems and test on the ImageNet (Deng et al., 2009) dataset. We defer more related details to Appendix B.1.

**Algorithms.** We compare the proposed Feature CP against the vanilla conformal baselines without further specified, which directly deploy conformal inference on the output space. For both methods, we use $\ell_\infty$-type non-conformity score, namely, $s(X, Y, \mu) = \|Y - \mu(X)\|_\infty$.

**Evaluation.** We adopt the following metrics to evaluate algorithmic empirical performance.

*Empirical coverage (effectiveness)* is the empirical probability that a test point falls into the predicted confidence band. A good predictive inference method should achieve empirical coverage slightly larger than $1 - \alpha$ for a given significance level $\alpha$. To calculate the coverage for Feature CP, we first apply Band Detection on the test point $(X', Y')$ to detect whether $Y'$ is in $\mathcal{C}_{1-\alpha}^{\text{fcp}}(X')$, and then calculate its average value to obtain the empirical coverage.

*Band length (efficiency).* Given the empirical coverage being larger than $1 - \alpha$, we hope the confidence band to be as short as possible. The band length should be compared under the regime of empirical coverage being larger than $1 - \alpha$, otherwise one can always set the confidence band to empty to get a zero band length. Since the explicit expression for confidence bands is intractable for the proposed algorithm, we could only derive an estimated band length via Band Estimation. Concretely, we

| Image | Ground Truth Label | Length |
| --- | --- | --- |

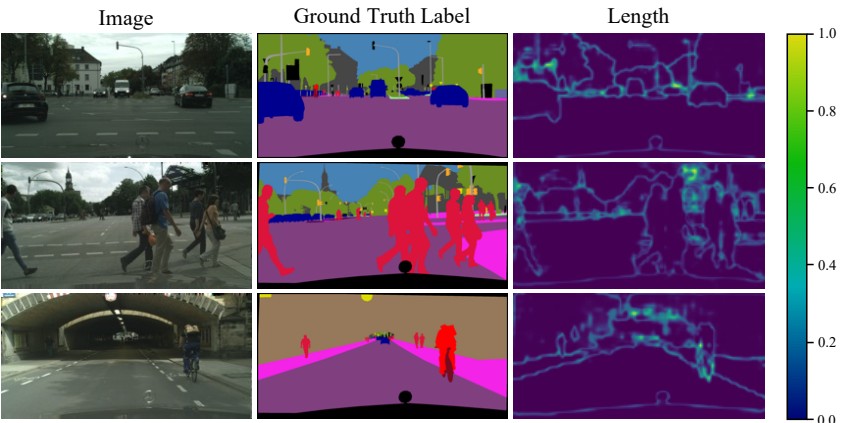

Figure 3: Visualization of Feature CP in image segmentation. The brightness of the pixels in the third column measures the uncertainty of Feature CP, namely the length of confidence bands. The algorithm is more uncertain in the brighter regions. For better visualization, we rescale the interval length to the range of $[0, 1]$. Feature CP is more uncertain in non-informative regions, which are object boundaries.

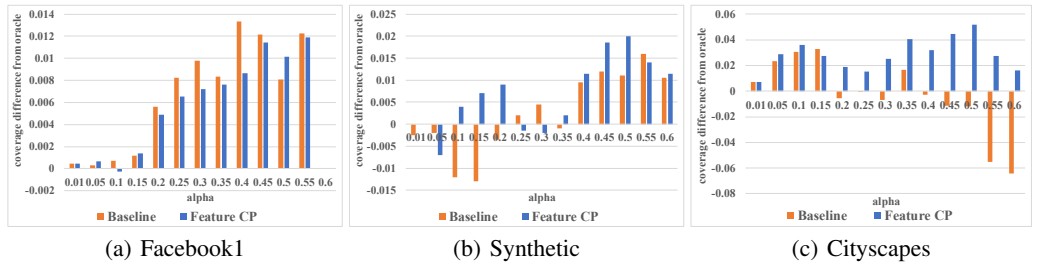

(a) Facebook1  (b) Synthetic  (c) Cityscapes

Figure 4: Empirical coverage under different confidence levels. For a good conformal prediction method, the $y$-axis (*i.e.*, empirical coverage minus $(1 - \alpha)$) should keep being above zero for different $\alpha$. These three figures above show that Feature CP generally performs better than the baseline, in the sense that this difference is above zero most of the time.

first use Band Estimation to estimate the confidence interval, which returns a band with explicit formulation, and then calculate the average length across each dimension.

We formulate the metrics as follows. Let $Y = (Y^{(1)}, \ldots, Y^{(d)}) \in \mathbb{R}^d$ denotes the high dimensional response and $\mathcal{C}(X) \subseteq \mathbb{R}^d$ denotes the obtained confidence interval, with length in each dimension forming a vector $|\mathcal{C}(X)| \in \mathbb{R}^d$. With the test set index being $\mathcal{I}_{\text{te}}$ and $[d] = \{1, \ldots, d\}$, we calculate the empirical coverage and band length respectively as

$$\frac{1}{|\mathcal{I}_{\text{te}}|} \sum_{i \in \mathcal{I}_{\text{te}}} \mathbb{I}(Y_i \in \mathcal{C}(X_i)), \quad \frac{1}{|\mathcal{I}_{\text{te}}|} \sum_{i \in \mathcal{I}_{\text{te}}} \frac{1}{d} \sum_{j \in [d]} |\mathcal{C}(X_i)|^{(j)}.$$

## 5.2 RESULTS AND DISCUSSION

**Effectiveness.** We summarize the empirical coverage in Figure 2 (one-dimension response) and Table 1 (multi-dimension response). As Theorem 5 illustrates, the empirical coverage of Feature CP all exceeds the confidence level $1 - \alpha$, indicating that Feature CP is effective. Besides, Figure 4 demonstrates that that the effectiveness holds with different significance levels $\alpha$. For simple benchmarks such as facebook1 and synthetic data, both methods achieve similar coverage due to the simplicity; while for the harder Cityscapes segmentation task, the proposed method outperforms the baseline under many confidence levels.

**Efficiency.** We summarize the confidence band in Figure 2 (one-dimension response) and Table 1 (multi-dimension response). The band length presented is an estimated version via Band Estimation.

Table 2: Feature CP outperform previous methods on large-scale classification tasks (ImageNet, $\alpha = 0.1$). We compare our results with APS (Romano et al., 2020a) and RAPS (Angelopoulos et al., 2021b). The results of baselines are taken from Angelopoulos et al. (2021b).

| METHOD | ACCURACY | | COVERAGE | | | LENGTH | | |
|---|---|---|---|---|---|---|---|---|
| MODEL | TOP-1 | TOP-5 | APS | RAPS | FEATURE CP | APS | RAPS | FEATURE CP |
| RESNET 18 | 0.698 | 0.891 | 0.900 | 0.900 | $0.902 \pm 0.0023$ | 16.2 | 4.43 | $\mathbf{3.82} \pm 0.18$ |
| RESNET 50 | 0.761 | 0.929 | 0.900 | 0.900 | $0.900 \pm 0.0047$ | 12.3 | 2.57 | $\mathbf{2.14} \pm 0.03$ |
| RESNET 101 | 0.774 | 0.936 | 0.900 | 0.900 | $0.900 \pm 0.0025$ | 10.7 | 2.25 | $\mathbf{2.17} \pm 0.01$ |
| RESNEXT 101 | 0.783 | 0.945 | 0.900 | 0.900 | $0.900 \pm 0.0030$ | 19.7 | 2.00 | $\mathbf{1.80} \pm 0.08$ |
| SHUFFLENET | 0.694 | 0.883 | 0.900 | 0.900 | $0.901 \pm 0.0019$ | 31.9 | 5.05 | $5.06 \pm 0.04$ |
| VGG 16 | 0.716 | 0.904 | 0.901 | 0.900 | $0.901 \pm 0.0047$ | 14.1 | 3.54 | $\mathbf{3.26} \pm 0.03$ |

Note that Feature CP outperforms the baseline in the sense that it achieves a shorter band length and thus a more efficient algorithm.

**Comparison to CQR.** The techniques proposed in this paper can be generalized to other conformal prediction techniques. As an example, we propose Feature CQR which is a feature-level generalized version of CQR, whose details are deferred to Appendix B.4. We display the comparison in Figure 2, where our method consistently outperforms CQR baseline by leveraging good representation. Besides, we evaluate the group coverage performance of CQR and Feature CQR in Appendix B.5, demonstrating that Feature CQR generally outperforms CQR in terms of condition coverage. One can also generalize the other existing techniques to feature versions, *e.g.*, Localized Conformal Prediction (Guan, 2019; Han et al., 2022).

**Extension to classification tasks.** The techniques in Feature CP can be generalized to classification tasks. We take the ImageNet classification dataset as an example. Table 2 shows the effectiveness of Feature CP in classification tasks. We experiment on various architectures and use the same pretrained weights as the baselines. We provide additional details in Appendix B.7.

**Ablation on the splitting point.** We demonstrate that the coverage is robust to the splitting point in both small neural networks (Table 9 in Appendix) and large neural networks (Table 10 in Appendix). One can also use standard cross-validation to choose the splitting point. Since the output space is one of the special feature layers, such techniques always generalize the scope of vanilla CP.

**Truthfulness.** We visualize the segmentation results in Figure 3, which illustrates that Feature CP returns large bands (light region) on the non-informative regions (object boundaries) and small bands (dark region) on the informative regions. We do not show baseline visualization results since they return the same band in each dimension for each sample, and therefore does not contain much information. We also evaluate the performance with weighted band length, defined in Appendix B.1.

**How does Feature CP benefit from a good deep representation?** Here we provide some intuition on the success of Feature CP algorithm: we claim it is the usage of good (deep) representation that enables Feature CP to achieve better predictive inference. To validate this hypothesis, we contrast Feature CP against the baseline with an unlearned neural network (whose feature is not semantic as desired). This "random" variant of Feature CP does not outperform its vanilla counterpart with the same neural network, which confirms our hypothesis. We defer the results to Table 6 and related discussion to Appendix B.3.

**More discussion.** We analyze the failure (*i.e.*, inefficient) reasons of vanilla CP in image segmentation task from the following two perspectives. Firstly, this paper aims to provide *provable coverage*, namely, the confidence band should cover the ground truth for each pixel. Since vanilla CP returns the same band for different samples, the loss is pretty large such that the returned interval is large enough to cover the ground truth. Secondly, an intuitive explanation relates to our usage of $\ell_\infty$ to form the non-conformity score during the training. We choose the infinity norm because reporting the total band length requires the band length in each dimension. As a result, the non-conformity score is large as long as there exists one pixel that does not fit well, contributing to an unsatisfying band for vanilla CP.

ACKNOWLEDGMENTS

This work has been partly supported by the Ministry of Science and Technology of the People's Republic of China, the 2030 Innovation Megaprojects "Program on New Generation Artificial Intelligence" (Grant No. 2021AAA0150000). This work is also supported by a grant from the Guoqiang Institute, Tsinghua University.

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

# Appendix

Section A provides the complete proofs, and Section B.1 provides experiment details.

## A    THEORETICAL PROOFS

We first show the formal version of Theorem 4 in Section A.1, and show its proof in Section A.2. Theorem 5 and Theorem 6 shows the effectiveness (empirical coverage) and the efficiency (band length) in Feature CP.

We additionally provide Theorem 9 (see Section A.3) and Theorem 15 (see Section A.4) to better validate our theorem, in terms of the length variance and convergence rate.

### A.1    THEORETICAL GUARANTEE

This section provides theoretical guarantees for Feature CP regarding coverage (effectiveness) and band length (efficiency), starting from additional notations.

**Notations.** Let $\mathcal{P}$ denote the population distribution. Let $\mathcal{D}_{\text{ca}} \sim \mathcal{P}^n$ denote the calibration set with sample size $n$ and sample index $\mathcal{I}_{\text{ca}}$, where we overload the notation $\mathcal{P}^n$ to denote the distribution of a set with samples drawn from distribution $\mathcal{P}$. Given the model $\hat{g} \circ \hat{f}$ with feature extractor $\hat{f}$ and prediction head $\hat{g}$, we assume $\hat{g}$ is continuous. We also overload the notation $Q_{1-\alpha}(V)$ to denote the $(1-\alpha)$-quantile of the set $V \cup \{\infty\}$. Besides, let $\mathbb{M}[\cdot]$ denote the mean of a set, and a set minus a real number denote the broadcast operation.

*Vanilla CP.* Let $V_{\mathcal{D}_{\text{ca}}}^o = \{v_i^o\}_{i \in \mathcal{I}_{\text{ca}}}$ denote the individual length in the output space for vanilla CP, given the calibration set $\mathcal{D}_{\text{ca}}$. Concretely, $v_i^o = 2|y_i - \hat{y}_i|$ where $y_i$ denotes the true response of sample $i$ and $\hat{y}_i$ denotes the corresponding prediction. Since vanilla CP returns band length with $1 - \alpha$ quantile of non-conformity score, the resulting average band length is derived by $Q_{1-\alpha}(V_{\mathcal{D}_{\text{ca}}}^o)$.

*Feature CP.* Let $V_{\mathcal{D}_{\text{ca}}}^f = \{v_i^f\}_{i \in \mathcal{I}_{\text{ca}}}$ be the individual length (or diameter in high dimensional cases) in the feature space for Feature CP, given the calibration set $\mathcal{D}_{\text{ca}}$. To characterize the band length in the output space, we define $\mathcal{H}(v, X)$ as the individual length on sample $X$ in the output space, given the length $v$ in the feature space, *i.e.*, $\mathcal{H}(v, X)$ represents the length of the set $\{\hat{g}(u) \in \mathbb{R} : \|u - \hat{f}(X)\| \leq v/2\}$. Due to the continuity assumption on function $\hat{g}$, the above set is always simply-connected. We here omit the dependency of prediction head $\hat{g}$ in $\mathcal{H}$ for simplicity. The resulting band length in Feature CP is denoted by $\mathbb{E}_{(X',Y') \sim \mathcal{P}} \mathcal{H}(Q_{1-\alpha}(V_{\mathcal{D}_{\text{ca}}}^f), X')$. Without abuse of notations, operating $\mathcal{H}$ on a dataset (*e.g.*, $\mathcal{H}(V_{\mathcal{D}_{\text{ca}}}^f, \mathcal{D}_{\text{ca}})$) means operating $\mathcal{H}$ on each data point $(v_i^f, X_i)$ in the set.

**Coverage guarantee.** We next provide theoretical guarantees for Feature CP in Theorem 5, which informally shows that under Assumption 1, the confidence band returned by Algorithm 3 is valid, meaning that the coverage is provably larger than $1 - \alpha$. We defer the whole proof to Appendix A.2.

**Theorem 5** (theoretical guarantee for Feature CP). *Under Assumption 1, for any $\alpha > 0$, the confidence band returned by Algorithm 3 satisfies:*

$$\mathbb{P}(Y' \in \mathcal{C}_{1-\alpha}^{fcp}(X')) \geq 1 - \alpha,$$

*where the probability is taken over the calibration fold and the testing point $(X', Y')$.*

**Length (efficiency) guarantee.** We next show in Theorem 6 that Feature CP is provably more efficient than the vanilla CP.

**Theorem 6** (Feature CP is provably more efficient). *Assume that the non-conformity score is in norm-type. For the operator $\mathcal{H}$, we assume a Holder assumption that there exist $\alpha > 0, L > 0$ such that $|\mathcal{H}(v, X) - \mathcal{H}(u, X)| \leq L|v - u|^\alpha$ for all $X$. Besides, we assume that there exists $\epsilon > 0, c > 0$, such that the feature space satisfies the following cubic conditions:*

1. **Length Preservation.** *Feature CP does not cost much loss in feature space in a quantile manner, namely, $\mathbb{E}_{\mathcal{D} \sim \mathcal{P}^n} Q_{1-\alpha}(\mathcal{H}(V_{\mathcal{D}}^f, \mathcal{D})) < \mathbb{E}_{\mathcal{D} \sim \mathcal{P}^n} Q_{1-\alpha}(V_{\mathcal{D}}^o) + \epsilon.$*

2. **Expansion.** *The operator $\mathcal{H}(v, X)$ expands the differences between individual length and their quantiles, namely, $L\mathbb{E}_{\mathcal{D}\sim\mathcal{P}^n}\mathbb{M}|Q_{1-\alpha}(V_{\mathcal{D}}^f) - V_{\mathcal{D}}^f|^\alpha < \mathbb{E}_{\mathcal{D}\sim\mathcal{P}^n}\mathbb{M}[Q_{1-\alpha}(\mathcal{H}(V_{\mathcal{D}}^f, \mathcal{D})) - \mathcal{H}(V_{\mathcal{D}}^f, \mathcal{D})] - \epsilon - 2\max\{L, 1\}(c/\sqrt{n})^{\min\{\alpha, 1\}}$.*

3. **Quantile Stability.** *Given a calibration set $\mathcal{D}_{ca}$, the quantile of the band length is stable in both feature space and output space, namely, $\mathbb{E}_{\mathcal{D}\sim\mathcal{P}^n}|Q_{1-\alpha}(V_{\mathcal{D}}^f) - Q_{1-\alpha}(V_{\mathcal{D}_{ca}}^f)| \le \frac{c}{\sqrt{n}}$ and $\mathbb{E}_{\mathcal{D}\sim\mathcal{P}^n}|Q_{1-\alpha}(V_{\mathcal{D}}^o) - Q_{1-\alpha}(V_{\mathcal{D}_{ca}}^o)| \le \frac{c}{\sqrt{n}}$.*

*Then Feature CP provably outperforms vanilla CP in terms of average band length, namely,*

$$\mathbb{E}\mathcal{H}(Q_{1-\alpha}(V_{\mathcal{D}_{ca}}^f), X') < Q_{1-\alpha}(V_{\mathcal{D}_{ca}}^o),$$

*where the expectation is taken over the calibration fold and the testing point $(X', Y')$.*

The cubic conditions used in Theorem 6 sketch the properties of feature space from different aspects. The first condition implies that the feature space is efficient for each individual, which holds when the band is generally not too large. The second condition is the core of the proof, which informally assumes that the difference between quantile and each individual is smaller in feature space. Therefore, conducting quantile operation would not harm the effectiveness (namely, step 4 in Algorithm 1 and step 4 in Algorithm 3), leading to the efficiency of Feature CP. The last condition helps generalize the results from the calibration set to the test set.

*Proof of Theorem 6.* We start the proof with Assumption 2, which claims that

$$L\mathbb{E}_{\mathcal{D}}\mathbb{M}|Q_{1-\alpha}(V_{\mathcal{D}}^f) - V_{\mathcal{D}}^f|^\alpha < \mathbb{E}_{\mathcal{D}}\mathbb{M}\left[Q_{1-\alpha}(\mathcal{H}(V_{\mathcal{D}}^f, \mathcal{D})) - \mathcal{H}(V_{\mathcal{D}}^f, \mathcal{D})\right]$$
$$- \epsilon - 2\max\{L, 1\}(c/\sqrt{n})^{\min\{\alpha, 1\}}.$$

We rewrite it as

$$\mathbb{E}_{\mathcal{D}}\mathbb{M}\mathcal{H}(V_{\mathcal{D}}^f, \mathcal{D}) < \mathbb{E}_{\mathcal{D}}Q_{1-\alpha}(\mathcal{H}(V_{\mathcal{D}}^f, \mathcal{D})) - \epsilon$$
$$- 2\max\{L, 1\}(c/\sqrt{n})^{\min\{\alpha, 1\}} - L\mathbb{E}_{\mathcal{D}}\mathbb{M}|Q_{1-\alpha}(V_{\mathcal{D}}^f) - V_{\mathcal{D}}^f|^\alpha.$$

Due to Holder condition, we have that $\mathbb{M}\mathcal{H}(Q_{1-\alpha}(V_{\mathcal{D}}^f), \mathcal{D}) < \mathbb{M}(\mathcal{H}(V_{\mathcal{D}}^f, \mathcal{D})) + L\mathbb{M}|Q_{1-\alpha}(V_{\mathcal{D}}^f) - V_{\mathcal{D}}^f|^\alpha$, therefore

$$\mathbb{E}_{\mathcal{D}}\mathbb{M}\left[\mathcal{H}(Q_{1-\alpha}(V_{\mathcal{D}}^f), \mathcal{D})\right] < \mathbb{E}_{\mathcal{D}}Q_{1-\alpha}(\mathcal{H}(V_{\mathcal{D}}^f, \mathcal{D})) - \epsilon - 2\max\{1, L\}[c/\sqrt{n}]^{\min\{1, \alpha\}}.$$

Therefore, due to assumption 1, we have that

$$\mathbb{E}_{\mathcal{D}}\mathbb{M}\mathcal{H}(Q_{1-\alpha}(V_{\mathcal{D}}^f), \mathcal{D}) < \mathbb{E}_{\mathcal{D}}Q_{1-\alpha}(V_{\mathcal{D}}^o) - 2\max\{1, L\}[c/\sqrt{n}]^{\min 1, \alpha}.$$

Besides, according to the quantile stability assumption, we have that $\mathbb{E}_{\mathcal{D}}|\mathbb{M}\mathcal{H}(Q_{1-\alpha}(V_{\mathcal{D}}^f), \mathcal{D}) - \mathbb{M}\mathcal{H}(Q_{1-\alpha}(V_{\mathcal{D}}^f), \mathcal{D})| \le L[c/\sqrt{n}]^\alpha$, and $\mathbb{E}_{\mathcal{D}}|Q_{1-\alpha}(V_{\mathcal{D}}^o) - Q_{1-\alpha}(V_{\mathcal{D}}^o)| \le c/\sqrt{n}$. Therefore,

$$\mathbb{E}\mathcal{H}(Q_{1-\alpha}(V_{\mathcal{D}_{ca}}^f), X')$$
$$= \mathbb{E}_{\mathcal{D}}\mathbb{M}\mathcal{H}(Q_{1-\alpha}(V_{\mathcal{D}_{ca}}^f), \mathcal{D})$$
$$< Q_{1-\alpha}(V_{\mathcal{D}_{ca}}^o) - 2\max\{1, L\}[c/\sqrt{n}]^{\min 1, \alpha} + L[c/\sqrt{n}]^\alpha + c/\sqrt{n}$$
$$< Q_{1-\alpha}(V_{\mathcal{D}_{ca}}^o).$$

□

### A.1.1 EXAMPLE FOR THEOREM 6

This section provides an example for Theorem 6. The key information is that Feature CP loses less efficiency when conducting the quantile step.

Table 3: A concrete example for the comparison between Feature CP and CP. Let $IL_o$, $IL_f$ denote the individual length in the feature and output space. Let $Q(\cdot)$ denote the quantile operator, and $\mathcal{H}(\cdot)$ denote the operator that calculates output space length given the feature space length. We remark that the average band length returned by Feature CP (3.1) outperforms that of vanilla CP (4.0).

| METHOD | VANILLA CP | | FEATURE CP | | | |
|---|---|---|---|---|---|---|
| SAMPLE | $IL_o$ | $Q(IL_o)$ | $IL_f$ | $\mathcal{H}(IL_f)$ | $Q(IL_f)$ | $\mathcal{H}(Q(IL_f))$ |
| A | 1.0 | 4.0 | 1.1 | 1.2 | 1.3 | 1.4 |
| B | 2.0 | 4.0 | 1.2 | 2.1 | 1.3 | 2.3 |
| C | 3.0 | 4.0 | 1.1 | 2.8 | 1.3 | 3.1 |
| D | 4.0 | 4.0 | 1.3 | 3.8 | 1.3 | 3.8 |
| E | 5.0 | 4.0 | 1.6 | 5.2 | 1.3 | 4.9 |
| QUANTILE | 4.0 | / | 1.3 | / | / | / |
| AVERAGE | / | **4.0** | / | / | / | **3.1** |

Assume the dataset has five samples labeled A, B, C, D, and E. When directly applying vanilla CP leads to individual length in the output space $IL_o$ as $1, 2, 3, 4, 5$, respectively. By taking $80\%$ quantile (namely, $\alpha = 0.2$), the final confidence band returned by vanilla CP ($Q(IL_o)$) would be $Q_{0.8}(\{1, 2, 3, 4, 5\}) = 4$. Note that for any sample, the returned band length would be $4$, and the final average band length is $4$.

We next consider Feature CP. We assume that the individual length in the feature space ($IL_f$) is $1.1, 1.2, 1.1, 1.3, 1.6$, respectively. Due to the expansion condition (cubic condition #2), the difference between $IL_f$ and $Q(IL_f)$ is smaller than that between $IL_o$ and $Q(IL_o)$. Therefore, the quantile step costs less in Feature CP. Since $IL_f$ is close to $Q(IL_f)$, their corresponding output length $\mathcal{H}(IL_f)$, $\mathcal{H}(Q(IL_f))$ are also close. Besides, to link conformal prediction and vanilla CP, the length preservation condition (cubic condition #1) ensures that $IL_o$ is close to $\mathcal{H}(IL_f)$. Therefore, the final average length $\mathbb{M}\mathcal{H}(Q(L_f))$ is close to the average length $\mathbb{M}IL_o$, which is better than $Q(IL_o)$ Finally, the quantile stability condition (cubic condition #3) generalizes the results from the calibration set to the test set.

## A.2 PROOF OF THEOREM 5

**Theorem 5** (theoretical guarantee for Feature CP). *Under Assumption 1, for any $\alpha > 0$, the confidence band returned by Algorithm 3 satisfies:*

$$\mathbb{P}(Y' \in \mathcal{C}^{fcp}_{1-\alpha}(X')) \geq 1 - \alpha,$$

*where the probability is taken over the calibration fold and the testing point $(X', Y')$.*

*Proof of Theorem 5.* The key to the proof is to derive the exchangeability of the non-conformity score, given that the data in the calibration fold and test fold are exchangeable (see Assumption 1).

For ease of notations, we denote the data points in the calibration fold and the test fold as $\mathcal{D}' = \{(X_i, Y_i)\}_{i \in [m]}$, where $m$ denotes the number of data points in both calibration fold and test fold. By Assumption 1, the data points in $\mathcal{D}'$ are exchangeable.

The proof can be split into three parts. The first step is to show that for any function independent of $\mathcal{D}'$, $h(X_i, Y_i)$ are exchangeable. The second step is to show that the proposed score function $s$ satisfies the above requirements. And the third step is to show the theoretical guarantee based on the exchangeability of the non-conformity score.

We next prove the first step: for any given function $h : \mathcal{X} \times \mathcal{Y} \to \mathbb{R}$ that is independent of data points in $\mathcal{D}'$, we have that $h(X_i, Y_i)$ are exchangeable. Specifically, its CDF $F_v$ and its perturbation CDF

$F_v^\pi$ is the same, given the training fold $\mathcal{D}_{\text{tr}}$.

$$
\begin{aligned}
&F_v(u_1, \ldots, u_n \mid \mathcal{D}_{\text{tr}}) \\
=&\mathbb{P}(h(X_1, Y_1) \leq u_1, \ldots, h(X_n, Y_n) \leq u_n \mid \mathcal{D}_{\text{tr}}) \\
=&\mathbb{P}((X_1, Y_1) \in \mathcal{C}_{h^{-1}}(u_1-), \ldots, (X_n, Y_n) \in \mathcal{C}_{h^{-1}}(u_n-) \mid \mathcal{D}_{\text{tr}}) \\
=&\mathbb{P}((X_{\pi(1)}, Y_{\pi(1)}) \in \mathcal{C}_{h^{-1}}(u_1-), \ldots, (X_{\pi(n)}, Y_{\pi(n)}) \in \mathcal{C}_{h^{-1}}(u_n-) \mid \mathcal{D}_{\text{tr}}) \\
=&\mathbb{P}(h(X_{\pi(1)}, Y_{\pi(1)}) \leq u_1, \ldots, h(X_{\pi(n)}, Y_{\pi(n)}) \leq u_n \mid \mathcal{D}_{\text{tr}}) \\
=&F_v^\pi(u_1, \ldots, u_n \mid \mathcal{D}_{\text{tr}}),
\end{aligned}
$$

where $\pi$ denotes a random perturbation, and $\mathcal{C}_{h^{-1}}(u-) = \{(X, Y) : h(X, Y) \leq u\}$.

The second step is to show that the proposed non-conformity score function (See Equation (3) and Algorithm 3) is independent of the dataset $\mathcal{D}'$. To show that, we note that the proposed score function $s$ in Equation (3) (we rewrite it in Equation (5)) is totally independent of dataset $\mathcal{D}'$, in that we only use the information of $\hat{f}$ and $\hat{g}$ which is dependent on the training fold $\mathcal{D}_{\text{tr}}$ instead of $\mathcal{D}'$.

$$
s(X, Y, \hat{g} \circ \hat{f}) = \inf_{v \in \{v : \hat{g}(v) = Y\}} \|v - \hat{f}(X)\|. \tag{5}
$$

Besides, note that when calculating the non-conformity score in Algorithm 3 for each testing data/calibration data, we do not access any information on the calibration folds for any other points. Therefore, the score function does not depend on the calibration fold or test fold. We finally remark that here we always state that the *score function $s$* does not depend on the calibration fold or test fold, but its realization $s(X, Y, \hat{g} \circ \hat{f})$ can depend on the two folds, if $(X, Y) \in \mathcal{D}'$. This does not contrast with the requirement in the first step.

Therefore, combining the two steps leads to a conclusion that the non-conformity scores on $\mathcal{D}'$ are exchangeable. Finally, following Lemma 1 in Tibshirani et al. (2019), the theoretical guarantee holds under the exchangeability of non-conformity scores. □

### A.3 LENGTH VARIANCE GUARANTEE

The next Theorem 9 demonstrates that the length returned by Feature CP would be individually different. Specifically, the variance for the length is lower bounded by a constant. The essential intuition is that, for a non-linear function $g$, the feature bands with the same length return different bands in output space. Before expressing the theorem, we first introduce a formal notation of length and other necessary assumptions. For ease of discussion, we define in Definition 7 a type of band length slightly different from the previous analysis. We assume $Y \in \mathbb{R}$ below, albeit our analysis can be directly extended to high-dimensional cases.

**Definition 7** (band length). *For a given feature $v$ and any perturbation $\tilde{v} \in \mathcal{C}_f(v) = \{\tilde{v} : \|\tilde{v} - v\| \leq Q\}$ in the feature band, we define the band length in the output space $L_o(v)$ as the maximum distance between predictor $g(v)$ and $g(\tilde{v})$, namely*

$$
L_o(v) \triangleq \max_{\tilde{v} \in \mathcal{C}_f(v)} |g(\tilde{v}) - g(v)|.
$$

Besides, we require Assumption 8, which is about the smoothness of the prediction head $g$.

**Assumption 8.** *Assume that the prediction head $g$ is second order derivative and $M$-smooth, namely, $\|\nabla^2 g(u)\| \leq M$ for all feasible $u$.*

The following Theorem 9 indicates that the variance of the band length is lower bounded, meaning that the bands given by Feature CP are individually different.

**Theorem 9.** *Under Assumption 8, if the band on the feature space is with radius $Q$, then the variance of band length on the output space satisfies:*

$$
\mathbb{E}\left[L_o - \mathbb{E}L_o\right]^2 / Q^2 \geq \mathbb{E}\left[\|\nabla g(v)\| - \mathbb{E}\|\nabla g(v)\|\right]^2 - MQ\mathbb{E}\|\nabla g(v)\|.
$$

From Theorem 9, the variance of the band length has a non-vacuous lower bound if

$$
\mathbb{E}[\|\nabla g(v)\| - \mathbb{E}\|\nabla g(v)\|]^2 > MQ \cdot \mathbb{E}\|\nabla g(v)\|. \tag{6}
$$

We next discuss the condition for Equation (6). For a linear function $g$, note that $\mathbb{E}[\|\nabla g(v)\| - \mathbb{E}\|\nabla g(v)\| = 0$ and $M = 0$, thus does not meet Equation (6). But for any other non-linear function $g$, we at least have $\mathbb{E}[\|\nabla g(v)\| - \mathbb{E}\|\nabla g(v)\|]^2 > 0$ and $M > 0$, and therefore there exists a term $Q$ such that Equation (6) holds. Hence, the band length in feature space must be individually different for a non-linear function $g$ and a small band length $Q$.

*Proof of Theorem 9.* We revisit the notation in the main text, where $v = f(X)$ denotes the feature, and $\mathcal{C}_f(v) = \{\tilde{v} : \|\tilde{v} - v\| \leq Q\}$ denotes the confidence band returned in feature space. By Taylor Expansion, for any given $\tilde{v} \in \mathcal{C}_f(v)$, there exists a $v'$ such that

$$g(\tilde{v}) - g(v) = \nabla g(v)(\tilde{v} - v) + 1/2(\tilde{v} - v)^\top \nabla^2 g(v')(\tilde{v} - v).$$

Due to Assumption 8, $\|\nabla^2 g(v')\| \leq M$. Therefore, for any $\tilde{v} \in \mathcal{C}_f(v)$

$$|1/2(\tilde{v} - v)^\top \nabla^2 g(v')(\tilde{v} - v)| \leq \frac{1}{2}MQ^2.$$

On the one hand, by Cauchy Schwarz inequality, we have

$$L_o = \max_{\tilde{v}} |g(\tilde{v}) - g(v)| \leq \|\nabla g(v)\|Q + \frac{1}{2}MQ^2.$$

On the other hand, by setting $\tilde{v} - v = Q\nabla g(v)/|\nabla g(v)|$, we have that

$$L_o = \max_{\tilde{v}} |g(\tilde{v}) - g(v)| \geq |g(v + Q\nabla g(v)/|\nabla g(v)|) - g(v)| = Q|\nabla g(v)| - 1/2MQ^2.$$

Therefore, we have that

$$|L_o - Q|\nabla g(v)|| \leq 1/2MQ^2.$$

We finally show the variance of the length, where the randomness is taken over the data $v$,

$$\mathbb{E}\left[[L_o - \mathbb{E}L_o]^2\right]$$
$$= \mathbb{E}\left[Q|\nabla g(v)| - \mathbb{E}Q|\nabla g(v)| + [L_o - Q|\nabla g(v)|] - \mathbb{E}[L_o - Q|\nabla g(v)|]\right]^2$$
$$= \mathbb{E}\left[Q|\nabla g(v)| - \mathbb{E}Q|\nabla g(v)|\right]^2 + \mathbb{E}\left[[L_o - Q|\nabla g(v)|] - \mathbb{E}[L_o - Q|\nabla g(v)|]\right]^2$$
$$\quad + 2\mathbb{E}\left[Q|\nabla g(v)| - \mathbb{E}Q|\nabla g(v)|\right]\left[(L_o - Q|\nabla g(v)|) - \mathbb{E}(L_o - Q|\nabla g(v)|)\right]$$
$$\geq Q^2\mathbb{E}\left[|\nabla g(v)| - \mathbb{E}|\nabla g(v)|\right]^2$$
$$\quad - 2Q\mathbb{E}\left[|\nabla g(v)| - \mathbb{E}|\nabla g(v)|\right]\left[(L_o - Q|\nabla g(v)|) - \mathbb{E}(L_o - Q|\nabla g(v)|)\right]$$
$$\geq Q^2\mathbb{E}\left[|\nabla g(v)| - \mathbb{E}|\nabla g(v)|\right]^2 - MQ^3\mathbb{E}\left[|\nabla g(v)| - \mathbb{E}|\nabla g(v)|\right].$$

Besides, note that $\mathbb{E}[|\nabla g(v)| - \mathbb{E}|\nabla g(v)|] \leq \mathbb{E}|\nabla g(v)|$. Therefore, we have that

$$\mathbb{E}\left[L_o - \mathbb{E}L_o\right]^2 / Q^2 \geq \mathbb{E}\left[|\nabla g(v)| - \mathbb{E}|\nabla g(v)|\right]^2 - MQ\mathbb{E}|\nabla g(v)|.$$

$\square$

## A.4 THEORETICAL CONVERGENCE RATE

In this section, we prove the theoretical convergence rate for the width. Specifically, we derive that when the number of samples in the calibration fold goes to infinity, the width for the testing point converges to a fixed value. Before we introduce the main theorem, we introduce some necessary definitions. Without further clarification, we follow the notations in the main text.

**Definition 10** (Precise Band). *We define the precise band as*

$$\mathcal{C}^{pre}_{1-\alpha} = \{g(v) : \|v - \hat{v}\| \leq Q_{1-\alpha}\}. \tag{7}$$

**Definition 11** (Precise Exact Band). *We define the exact precise band as*

$$\bar{\mathcal{C}}^{pre}_{1-\alpha} = \{g(v) : \|v - \hat{v}\| \leq \bar{Q}_{1-\alpha}\}, \tag{8}$$

*where $\bar{Q}_{1-\alpha}$ denotes the exact value such that*

$$\mathbb{P}(\exists v : \|v - \hat{v}\| \leq \bar{Q}_{1-\alpha}, g(v) = y) = 1 - \alpha. \tag{9}$$

Our goal is to prove that the band length (volume) of $\mathcal{C}_{1-\alpha}^{\text{pre}}$ (denoted by $\mathcal{V}(\mathcal{C}_{1-\alpha}^{\text{pre}})$) converges to $\mathcal{V}(\bar{\mathcal{C}}_{1-\alpha}^{\text{pre}})$. We assume that the prediction head and the quantile function are both Lipschitz in Assumption 12 and Assumption 13.

**Assumption 12** (Lipschitz for Prediction Head). *Assume that for any $v, v'$, we have*

$$\|g(v) - g(v')\| \leq L_1 \|v - v'\|.$$

**Assumption 13** (Lipschitz for Inverse Quantile Function). *Denote the quantile function as*

$$Quantile(Q_u) = \mathbb{P}(\exists v : \|v - \hat{v}\| \leq Q_u, g(v) = y) = u.$$

*We assume that its inverse function is $L_2$-Lipschitz, that is to say,*

$$\|Quantile^{-1}(u) - Quantile^{-1}(u')\| \leq L_2 \|u - u'\|.$$

Besides, we assume that the region of $\bar{\mathcal{C}}_{1-\alpha}^{\text{pre}}$ has benign blow-up.

**Assumption 14** (Benign Blow-up). *Assume that $\bar{\mathcal{C}}_{1-\alpha}^{pre}$ has benign blow-up, that is to say, for the blow-up set $\mathcal{C}_{1-\alpha}^{pre}(\epsilon) = \{v : \exists u \in \bar{\mathcal{C}}_{1-\alpha}^{pre}, \|u - v\| \leq \epsilon\}$, we have*

$$\|\mathcal{V}(\mathcal{C}_{1-\alpha}^{pre}(\epsilon)) - \mathcal{V}(\bar{\mathcal{C}}_{1-\alpha}^{pre})\| \leq c\epsilon,$$

*where $c$ denotes a constant independent of $n$.*

In the one-dimensional case $Y \in \mathbb{R}$, Assumption 14 easily holds. For the high-dimensional cases, such a bound usually requires that $c$ depends on the dimension $d$.

**Theorem 15** (Convergence Rate). *Assume that the non-conformity scores in the calibration fold have no ties. Under Assumption 12, Assumption 13 and Assumption 14, we have that*

$$\|\mathcal{V}(\mathcal{C}_{1-\alpha}^{pre}) - \mathcal{V}(\bar{\mathcal{C}}_{1-\alpha}^{pre})\| \leq cL_1 L_2 \frac{1}{n}.$$

*Proof.* Firstly, as derived in Romano et al. (2019), when the non-conformity score in the calibration fold has no ties (the probability is zero), we have

$$\mathbb{P}(\exists v : \|v - \hat{v}\| \leq Q_{1-\alpha}, g(v) = y) \in (1 - \alpha, 1 - \alpha + 1/n), \tag{10}$$

where $v, \hat{v}, Q_{1-\alpha}$ denotes the surrogate feature, the trained feature, and the quantile value in Algorithm 3, respectively.

By Assumption 13 that the inverse quantile function is $L_1$-Lipschitz around $1 - \alpha$, we have

$$\|\bar{Q}_{1-\alpha} - Q_{1-\alpha}\| \leq L_2 \frac{1}{n}.$$

Therefore, for any $u \in \mathcal{C}_{1-\alpha}^{\text{pre}}$, there *exists* $u' \in \bar{\mathcal{C}}_{1-\alpha}^{\text{pre}}$ such that

$$\|u - u'\| \triangleq \|g(v) - g(v')\| \leq L_2 \|v - v'\| \leq L_1 L_2 \frac{1}{n}. \tag{11}$$

We note that bounding $\|v - v'\|$ requires that the region of $v, v'$ are both balls, and therefore one can select $v'$ as the point with the smallest distance to $v$. Since the region of $\bar{\mathcal{C}}_{1-\alpha}^{\text{pre}}$ has benign blow-up, we have that

$$\mathcal{V}(\mathcal{C}_{1-\alpha}^{\text{pre}}) \leq \mathcal{V}(\bar{\mathcal{C}}_{1-\alpha}^{\text{pre}}) + cL_1 L_2 \frac{1}{n}.$$

Besides, the following equation naturally holds due to Equation (10).

$$\mathcal{V}(\mathcal{C}_{1-\alpha}^{\text{pre}}) \geq \mathcal{V}(\bar{\mathcal{C}}_{1-\alpha}^{\text{pre}}).$$

Therefore, we conclude with the following inequality,

$$\|\mathcal{V}(\mathcal{C}_{1-\alpha}^{\text{pre}}) - \mathcal{V}(\bar{\mathcal{C}}_{1-\alpha}^{\text{pre}})\| \leq cL_1 L_2 \frac{1}{n}.$$

Therefore, as the sample size in the calibration fold goes to infinity, the length of the trained band converges to $\mathcal{V}(\bar{\mathcal{C}}_{1-\alpha}^{\text{pre}})$.

$\square$

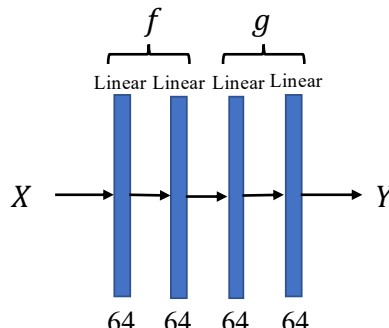

Figure 5: The model architecture of the uni-dimensional and synthetic multi-dimensional target regression experiments. The dropout layers are omitted.

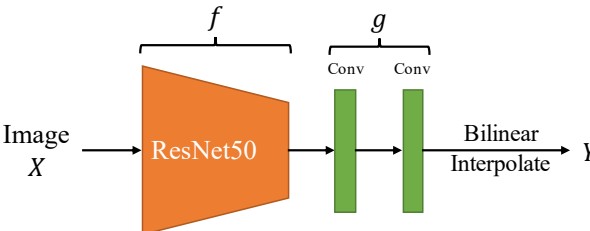

Figure 6: The model architecture of the semantic segmentation experiment.

## B    EXPERIMENTAL DETAILS

Section B.1 introduces the omitted experimental details. Section B.2 provide experimental evidence to validate cubic conditions. Section B.3 shows that Feature CP performs similarly to vanilla CP for untrained neural networks, validating that Feature CP works due to semantic information trained in feature space. Section B.4 introduces Feature CQR which applies feature-level techniques on CQR and Section B.5 reports the corresponding group coverage. Finally, Section B.9 provides other additional experiments omitted in the main text.

### B.1    EXPERIMENTAL DETAILS

**Model Architecture.** The model architecture of the uni-dimensional and synthetic multi-dimensional target regression task is shown in Figure 5. The feature function $f$ and prediction head $g$ includes two linear layers, respectively. Moreover, the model architecture of the FCN used in the semantic segmentation experiment is shown in Figure 6, which follows the official implementation of PyTorch. The batch normalization and dropout layers are omitted in the figure. We use the ResNet50 backbone as $f$ and take two convolution layers as $g$. We select the Layer4 output of ResNet50 as our surrogate feature $v$.

**Training protocols.** In the unidimensional and synthetic dimensional target regression experiments, we randomly divide the dataset into training, calibration, and test sets with the proportion $2:2:1$. As for the semantic segmentation experiment, because the labels of the pre-divided test set are not accessible, we re-split the training, calibration, and test sets randomly on the original training set of Cityscapes. We remove the class $0$ (unlabeled) from the labels during calibration and testing, and use the weighted mean square error as the training objective where the class weights are adopted from Paszke et al. (2016).

**Band Estimation.** In estimating the band length, we deploy Band Estimation based on the score calculated by Algorithm 2. In our experiment, we choose the number of steps $M$ in Algorithm 2 via cross-validation.

Table 4: Comparison between CP and Feature CP.

| METHOD | CP | | FEATURE CP | |
|---|---|---|---|---|
| DATASET | COVERAGE | LENGTH | COVERAGE | LENGTH |
| COMMUNITY | 89.82 ±0.95 | 1.99±0.09 | 89.62±0.83 | 1.99±0.19 |
| FACEBOOK1 | 90.11 ±0.33 | 3.17 ±0.59 | 90.07 ±0.32 | **2.08** ±0.17 |
| FACEBOOK2 | 89.99 ±0.18 | 2.72 ±0.37 | 89.98 ±0.17 | **1.97** ±0.17 |
| MEPS19 | 90.51 ±0.21 | 4.25 ±0.21 | 90.55 ±0.17 | **3.58** ±0.35 |
| MEPS20 | 89.80 ±0.61 | 4.17 ±0.36 | 89.76 ±0.61 | **3.37** ±0.38 |
| MEPS21 | 89.92 ±0.54 | 4.67 ±0.30 | 89.94 ±0.54 | 3.93 ±0.54 |
| STAR | 90.30 ±1.17 | 0.41 ±0.02 | 90.35 ±0.99 | 0.41 ±0.03 |
| BIO | 90.27 ±0.26 | 1.97 ±0.01 | 90.20 ±0.29 | **1.87** ±0.04 |
| BLOG | 90.08 ±0.27 | 3.32 ±0.38 | 90.06 ±0.33 | 2.81 ±0.40 |
| BIKE | 89.65 ±0.87 | 1.91 ±0.04 | 89.61 ±0.69 | **1.79** ±0.08 |

**Randomness.** We train each model five times with different random seeds and report the mean and standard deviation value across all the runs as the experimental results (as shown in Figure 2 and Table 1).

**Details of transforming segmentation classification problem into a regression task.** The original semantic segmentation problem is to fit the one-hot label $y$ whose size is $(C, W, H)$ via logistic regression, where $C$ is the number of the classes, $W$ and $H$ are the width and height of the image. We use Gaussian Blur to smooth the values in each channel of $y$. At this time, the smoothed label $\tilde{y}$ ranges from 0 to 1. Then, we use the double log trick to convert the label space from $[0, 1]$ to $(-\infty, \infty)$, *i.e.*, $\dot{y} = \log(-\log(\tilde{y}))$. Finally, we use mean square error loss to fit $\dot{y}$.

**Definition of weighted length.** We formulate the weighted length as

$$\text{weighted length} = \frac{1}{|\mathcal{I}_{\text{te}}|} \sum_{i \in \mathcal{I}_{\text{te}}} \sum_{j \in [d]} w_i^{(j)} |\mathcal{C}(X_i)|^{(j)},$$

where $w_i^{(j)}$ is the corresponding weight in each dimension. We remark that although the formulation of $w_i^{(j)}$ is usually sample-dependent, we omit the dependency of the sample and denote it by $w^{(j)}$ when the context is clear. We next show how to define $w^{(j)}$ in practice.

Generally speaking, we hope that $w^{(j)}$ is large when being informative (*i.e.*, in non-boundary regions). Therefore, for the $j$-th pixel after Gaussian Blur whose value is $Y^{(j)} \in [0, 1]$, its corresponding weight is defined as

$$w^{(j)} = \frac{|2Y^{(j)} - 1|}{W} \in [0, 1],$$

where $W = \sum_j |2Y^{(j)} - 1|$ is a scaling factor.

At a colloquial level, $w^{(j)}$ is close to 1 if $Y^{(j)}$ is close to 0 or 1. In this case, $Y^{(j)}$ being close to 0 or 1 means that the pixel is far from the boundary region. Therefore, the weight indicates the degree to which a pixel is being informative (not in object boundary regions).

**Calibration details.** During calibration, to get the best value for the number of steps $M$, we take a subset (one-fifth) of the calibration set as the additional validation set. We calculate the non-conformity score on the rest of the calibration set with various values of step $M$ and then evaluate on the validation set to get the best $M$ whose coverage is just over $1 - \alpha$. The final trained surrogate feature $v$ is close to the true feature because $\hat{g}(v)$ is sufficiently close to the ground truth $Y$. In practice, the surrogate feature after optimization satisfies $\frac{\|\hat{g}(v) - Y\|^2}{\|Y\|^2} < 1\%$.

**Comparison between Feature CP and Vanilla CP.** We next present the specific statistics of figure 2 in Table 4 and Table 7.

Table 5: Validate cubic conditions.

| SPACE | FEATURE SPACE | OUTPUT SPACE |
|---|---|---|
| METRIC | $\mathbb{M}\|Q_{1-\alpha}V_{\mathcal{D}_{\text{CA}}}^{f} - V_{\mathcal{D}_{\text{CA}}}^{f}\|$ | $\mathbb{M}[Q_{1-\alpha}\mathcal{H}(V_{\mathcal{D}_{\text{CA}}}^{f}) - \mathcal{H}(V_{\mathcal{D}_{\text{CA}}}^{f})]$ |
| COMMUNITY | $0.1150 \pm 0.0290$ | $0.8073 \pm 0.1450$ |
| FACEBOOK1 | $0.2491 \pm 0.0391$ | $1.8950 \pm 0.2058$ |
| FACEBOOK2 | $0.2387 \pm 0.0960$ | $1.7918 \pm 0.5220$ |
| MEPS19 | $0.2403 \pm 0.0161$ | $1.7511 \pm 0.1150$ |
| MEPS20 | $0.2485 \pm 0.0571$ | $1.7936 \pm 0.3532$ |
| MEPS21 | $0.2528 \pm 0.0488$ | $1.8686 \pm 0.3350$ |
| STAR | $0.0230 \pm 0.0025$ | $0.1605 \pm 0.0123$ |
| BIO | $0.1051 \pm 0.0056$ | $0.8509 \pm 0.0368$ |
| BLOG | $0.3537 \pm 0.1135$ | $2.3769 \pm 0.5421$ |
| BIKE | $0.0921 \pm 0.0058$ | $0.7759 \pm 0.0469$ |

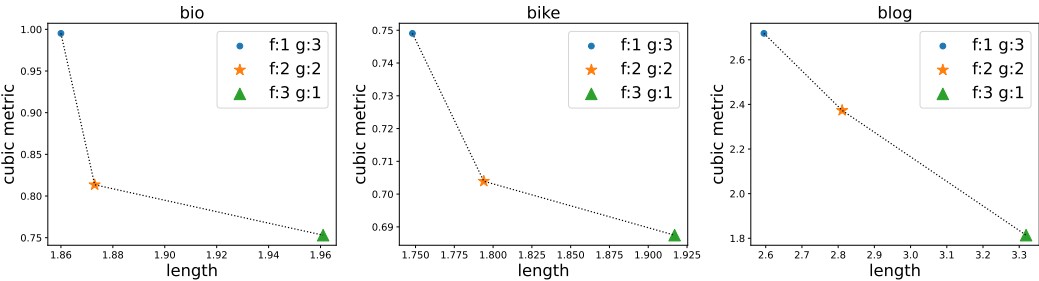

Figure 7: Length v.s. cubic metric. Larger cubic metric implies better efficiency (shorter band).

## B.2 CERTIFYING CUBIC CONDITIONS

In this section, we validate the cubic conditions. The most important component for the cubic condition is Condition 2, which claims that conducting the quantile step would not hurt much efficiency. We next provide experiment results in Table 5 on comparing the average distance between each sample to their quantile in feature space $\mathbb{M}\|Q_{1-\alpha}V_{\mathcal{D}_{\text{ca}}}^{f} - V_{\mathcal{D}_{\text{ca}}}^{f}\|$ and in output space $\mathbb{M}[Q_{1-\alpha}\mathcal{H}(V_{\mathcal{D}_{\text{ca}}}^{f}, \mathcal{D}_{\text{ca}}) - \mathcal{H}(V_{\mathcal{D}_{\text{ca}}}^{f}, \mathcal{D}_{\text{ca}})]$. We here take $\alpha = 1$ for simplicity. The significant gap in Table 5 validates that the distance in feature space is significantly smaller than that in output space, although we did not consider the Lipschitz factor $L$ for computational simplicity.

Besides, we plot the relationship between the efficiency (band length) v.s. cubic metric in Figure 7. Specifically, cubic metric here represents the core statement in cubic condition (statement 2), which implies a metric form like $\mathbb{M}\|Q_{1-\alpha}V_{\mathcal{D}_{\text{ca}}}^{f} - V_{\mathcal{D}_{\text{ca}}}^{f}\|$. The results are shown in Figure 7.

## B.3 FEATURE CP WORKS DUE TO SEMANTIC INFORMATION IN FEATURE SPACE

Experiment results illustrate that feature-level techniques improve the efficiency of conformal prediction methods (*e.g.*, Feature CP vs. CP, Feature CQR vs. CQR). We claim that *exploiting the semantic information in feature space is the key to our algorithm*. Different from most existing conformal prediction algorithms, which regard the base model as a black-box mode, feature-level operations allow seeing the training process via the trained feature. This is novel and greatly broadens the scope of conformal prediction algorithms. For a well-trained base model, feature-level techniques improve efficiency by utilizing the powerful feature embedding abilities of well-trained neural networks.

In contrast, if the base model is untrained with random initialization (whose representation space does not have semantic meaning), Feature CP returns a similar band length as the baseline (see Table 6). This validates the hypothesis that Feature CP's success lies in leveraging the inductive bias of deep representation learning. Fortunately, realistic machine learning models usually contain meaningful information in the feature space, enabling Feature CP to perform well.

Table 6: **Untrained base model comparison between conformal prediction and Feature CP.** The base model is randomly initialized but not trained with the training fold. Experiment results show that Feature CP cannot outperform vanilla CP if the base model is not well-trained.

| METHOD | VANILLA CP | | FEATURE CP | |
|---|---|---|---|---|
| DATASET | COVERAGE | LENGTH | COVERAGE | LENGTH |
| COMMUNITY | $90.28 \pm 1.70$ | $\mathbf{4.85} \pm 0.22$ | $90.68 \pm 1.33$ | $4.92 \pm 0.77$ |
| FACEBOOK1 | $90.15 \pm 0.15$ | $3.42 \pm 0.25$ | $90.16 \pm 0.12$ | $\mathbf{3.20} \pm 0.50$ |
| FACEBOOK2 | $90.17 \pm 0.11$ | $3.51 \pm 0.26$ | $90.12 \pm 0.14$ | $\mathbf{3.34} \pm 0.39$ |
| MEPS19 | $90.81 \pm 0.46$ | $\mathbf{4.02} \pm 0.16$ | $90.86 \pm 0.30$ | $4.22 \pm 0.48$ |
| MEPS20 | $90.10 \pm 0.60$ | $4.10 \pm 0.28$ | $90.28 \pm 0.46$ | $\mathbf{4.02} \pm 0.41$ |
| MEPS21 | $89.78 \pm 0.44$ | $4.08 \pm 0.16$ | $89.85 \pm 0.58$ | $\mathbf{3.81} \pm 0.32$ |
| STAR | $90.07 \pm 0.77$ | $\mathbf{2.23} \pm 0.18$ | $89.47 \pm 1.84$ | $2.24 \pm 0.40$ |
| BIO | $90.06 \pm 0.19$ | $\mathbf{4.25} \pm 0.11$ | $90.11 \pm 0.07$ | $4.44 \pm 0.74$ |
| BLOG | $90.13 \pm 0.34$ | $\mathbf{2.41} \pm 0.15$ | $90.16 \pm 0.26$ | $2.58 \pm 0.49$ |
| BIKE | $89.53 \pm 0.78$ | $4.65 \pm 0.15$ | $89.61 \pm 0.86$ | $\mathbf{4.13} \pm 0.38$ |

---

**Algorithm 4** Feature Conformalized Quantile Regression (Feature CQR)

---

**Require:** Level $\alpha$, dataset $\mathcal{D} = \{(X_i, Y_i)\}_{i \in \mathcal{I}}$, test point $X'$;
1: Randomly split the dataset $\mathcal{D}$ into a training fold $\mathcal{D}_{\text{tr}} \triangleq (X_i, Y_i)_{i \in \mathcal{I}_{\text{tr}}}$ together with a calibration fold $\mathcal{D}_{\text{ca}} \triangleq (X_i, Y_i)_{i \in \mathcal{I}_{\text{ca}}}$;
2: Train a base machine learning model $\hat{g}^{\text{lo}} \circ \hat{f}^{\text{lo}}(\cdot)$ and $\hat{g}^{\text{hi}} \circ \hat{f}^{\text{hi}}(\cdot)$ using $\mathcal{D}_{\text{tr}}$ to estimate the quantile of response $Y_i$, which returns $[\hat{Y}_i^{\text{lo}}, \hat{Y}_i^{\text{hi}}]$;
3: For each $i \in \mathcal{I}_{\text{ca}}$, calculate the index $c_i^{\text{lo}} = \mathbb{I}(\hat{Y}_i^{\text{lo}} \leq Y)$ and $c_i^{\text{hi}} = \mathbb{I}(\hat{Y}_i^{\text{hi}} \geq Y)$;
4: For each $i \in \mathcal{I}_{\text{ca}}$, calculate the non-conformity score $V_i^{\text{lo}} = \tilde{V}_i^{\text{lo}} c_i^{\text{lo}}$ where $\tilde{V}_i^{\text{lo}}$ is derived on the lower bound function with Algorithm 2;
5: Calculate the $(1 - \alpha)$-th quantile $Q_{1-\alpha}^{\text{lo}}$ of the distribution $\frac{1}{|\mathcal{I}_{\text{ca}}|+1} \sum_{i \in \mathcal{I}_{ca}} \delta_{V_i^{\text{lo}}} + \delta_{\infty}$;
6: Apply Band Estimation on test data feature $\hat{f}^{\text{lo}}(X')$ with perturbation $Q_{1-\alpha}^{\text{lo}}$ and prediction head $\hat{g}^{\text{lo}}$, which returns $[\mathcal{C}_0^{\text{lo}}, \mathcal{C}_1^{\text{lo}}]$;
7: Apply STEP 4-6 similarly with higher quantile, which returns $[\mathcal{C}_0^{\text{hi}}, \mathcal{C}_1^{\text{hi}}]$;
8: Derive $\mathcal{C}_{1-\alpha}^{\text{fcqr}}(X)$ based on Equation (12);
**Ensure:** $\mathcal{C}_{1-\alpha}^{\text{fcqr}}(X)$.

---

### B.4 FEATURE CONFORMALIZED QUANTILE REGRESSION

In this section, we show that feature-level techniques are pretty general in that they can be applied to most of the existing conformal prediction algorithms. Specifically, We take Conformalized Quantile Regression (CQR, Romano et al. (2019)) as an example and propose Feature-level Conformalized Quantile Regression (Feature CQR). The core idea is similar to Feature CP (See Algorithm 3), where we conduct calibration steps in the feature space. We summarize the Feature CQR algorithm in Algorithm 4.

Similar to CQR, Algorithm 4 also considers the one-dimension case where $Y \in \mathbb{R}$. We next discuss the steps in Algorithm 4. Firstly, different from Feature CP, Feature CQR follows the idea of CQR that the non-conformity score can be negative (see Step 4). Such negative scores help reduce the band length, which improves efficiency. This is achieved by the index calculated in Step 5[3]. Generally, if the predicted value is larger than the true value $\hat{Y}_i^{\text{lo}} > Y_i$, we need to adjust $\hat{Y}_i^{\text{lo}}$ to be smaller, and vice visa. Step 8 follows the adjustment, where we summarize the criterion in Equation (12), given the two band $[\mathcal{C}_0^{\text{lo}}, \mathcal{C}_1^{\text{lo}}]$ and $[\mathcal{C}_0^{\text{hi}}, \mathcal{C}_1^{\text{hi}}]$.

---

[3]Here, we set $\mathbb{I} = \pm 1$ for simplicity

Table 7: Comparison between CQR and Feature CQR. Feature CQR achieves better efficiency while maintaining effectiveness.

| METHOD | CQR | | FEATURE CQR | |
| DATASET | COVERAGE | LENGTH | COVERAGE | LENGTH |
|---|---|---|---|---|
| COMMUNITY | $90.33 \pm 1.36$ | $1.60 \pm 0.08$ | $90.23 \pm 1.65$ | $\mathbf{1.23} \pm 0.15$ |
| FACEBOOK1 | $89.94 \pm 0.23$ | $1.15 \pm 0.04$ | $92.00 \pm 0.20$ | $\mathbf{1.00} \pm 0.04$ |
| FACEBOOK2 | $89.99 \pm 0.03$ | $1.25 \pm 0.08$ | $92.19 \pm 0.17$ | $\mathbf{1.08} \pm 0.06$ |
| MEPS19 | $90.26 \pm 0.38$ | $2.41 \pm 0.26$ | $91.25 \pm 0.43$ | $\mathbf{1.48} \pm 0.19$ |
| MEPS20 | $89.78 \pm 0.60$ | $2.47 \pm 0.10$ | $90.9 \pm 0.39$ | $\mathbf{1.34} \pm 0.35$ |
| MEPS21 | $89.52 \pm 0.32$ | $2.26 \pm 0.20$ | $90.2 \pm 0.53$ | $\mathbf{1.69} \pm 0.20$ |
| STAR | $90.99 \pm 1.08$ | $0.20 \pm 0.00$ | $89.88 \pm 0.33$ | $\mathbf{0.13} \pm 0.01$ |
| BIO | $90.09 \pm 0.36$ | $1.39 \pm 0.01$ | $89.88 \pm 0.27$ | $\mathbf{1.22} \pm 0.02$ |
| BLOG | $90.15 \pm 0.15$ | $1.47 \pm 0.05$ | $91.49 \pm 0.26$ | $\mathbf{0.89} \pm 0.03$ |
| BIKE | $89.38 \pm 0.25$ | $0.58 \pm 0.01$ | $89.95 \pm 0.98$ | $\mathbf{0.38} \pm 0.02$ |

$$\text{if } c_i^{\text{lo}} < 0, c_i^{\text{hi}} < 0, \text{ return } \mathcal{C}_{1-\alpha}^{\text{fcqr}}(X) = [\mathcal{C}_0^{\text{lo}}, \mathcal{C}_1^{\text{hi}}];$$
$$\text{if } c_i^{\text{lo}} < 0, c_i^{\text{hi}} > 0, \text{ return } \mathcal{C}_{1-\alpha}^{\text{fcqr}}(X) = [\mathcal{C}_0^{\text{lo}}, \mathcal{C}_0^{\text{hi}}];$$
$$\text{if } c_i^{\text{lo}} > 0, c_i^{\text{hi}} < 0, \text{ return } \mathcal{C}_{1-\alpha}^{\text{fcqr}}(X) = [\mathcal{C}_1^{\text{lo}}, \mathcal{C}_1^{\text{hi}}];$$
$$\text{if } c_i^{\text{lo}} > 0, c_i^{\text{hi}} > 0, \text{ return } \mathcal{C}_{1-\alpha}^{\text{fcqr}}(X) = [\mathcal{C}_1^{\text{lo}}, \mathcal{C}_0^{\text{hi}}].$$
$$(12)$$

Similar to Feature CP, we need a Band Estimation step to approximate the band length used in Step 6. One can change it into Band Detection if necessary. Different from Feature CP where Band Estimation always returns the upper bound of the band, Feature CQR can only approximate it. We conduct experiments to show that this approximation does not lose effectiveness since the coverage is always approximate to $1 - \alpha$. Besides, different from CQR, which considers adjusting the upper and lower with the same value, we adjust them separately, which is more flexible in practice (see Step 7).

We summarize the experiments result in Table 7. Feature CQR achieves better efficiency while maintaining effectiveness. Here we provide 90% confidence band using five repeated experiments with different random seeds.

## B.5 GROUP COVERAGE FOR FEATURE CONFORMALIZED QUANTILE REGRESSION

This section introduces the group coverage returned by feature-level techniques, which implies the performance conditional coverage, namely $\mathbb{P}(Y \in \mathcal{C}(X)|X)$. Specifically, we split the test set into three groups according to their response values, and report the minimum coverage over each group.

We remark that the group coverage of feature-level conformal prediction stems from its vanilla version. That is to say, when the vanilla version has a satisfying group coverage, its feature-level version may also return a relatively satisfying group coverage. Therefore, we did not provide Feature CP here because vanilla CP cannot return a good group coverage.

We summarize the experiment results in Table 8. Although we did not provide a theoretical guarantee for group coverage, Feature CQR still outperforms vanilla CQR in various datasets in terms of group coverage. Among ten datasets, Feature CQR outperforms vanilla CQR in four datasets, and is comparable with vanilla CQR in five datasets. Although the advantage is not universal, improving group coverage via feature-level techniques is still possible.

We note that there is still one dataset where vanilla CQR outperforms Feature CQR. We attribute the possible failure reason of Feature CQR on the dataset FACEBOOK2 to the failure of base models. As stated in Section B.3, Feature CQR only works when the base model is well-trained. However, when grouping according to the returned values, it is possible that there exists one group that is not well-trained during the training process. This may cause the failure of Feature CQR on the dataset FACEBOOK2.

Table 8: Comparison between Feature CQR and CQR in terms of group coverage.

| GROUP COVERAGE | FEATURE CQR | VANILLA CQR |
|---|---|---|
| COMMUNITY | $76.05 \pm 5.13$ | $78.77 \pm 4.17$ |
| FACEBOOK1 | $65.52 \pm 0.95$ | $66.68 \pm 1.69$ |
| FACEBOOK2 | $65.66 \pm 1.41$ | $\mathbf{70.78} \pm 1.39$ |
| MEPS19 | $\mathbf{76.67} \pm 2.17$ | $71.26 \pm 1.23$ |
| MEPS20 | $\mathbf{77.70} \pm 0.90$ | $71.26 \pm 3.20$ |
| MEPS21 | $74.71 \pm 2.36$ | $70.74 \pm 1.83$ |
| STAR | $84.62 \pm 2.77$ | $82.20 \pm 5.72$ |
| BIO | $\mathbf{84.80} \pm 1.05$ | $80.03 \pm 1.48$ |
| BLOG | $\mathbf{59.43} \pm 0.60$ | $49.10 \pm 0.54$ |
| BIKE | $81.07 \pm 1.65$ | $78.22 \pm 2.44$ |

Table 9: Ablation study of the number of layers in $f$ and $g$ ($\alpha = 0.1$) in unidimensional tasks, where the default setting is $f : 2, g : 2$.

| DATASET | BIO | | BIKE | | BLOG | |
|---|---|---|---|---|---|---|
| METHOD | COVERAGE | LENGTH | COVERAGE | LENGTH | COVERAGE | LENGTH |
| $f : 2 \quad g : 2$ | $90.20 \pm 0.39$ | $1.873 \pm 0.06$ | $89.61 \pm 0.94$ | $1.794 \pm 0.11$ | $90.06 \pm 0.44$ | $2.811 \pm 0.54$ |
| $f : 3 \quad g : 1$ | $90.24 \pm 0.32$ | $1.961 \pm 0.02$ | $89.72 \pm 1.10$ | $1.917 \pm 0.14$ | $90.16 \pm 0.34$ | $3.319 \pm 0.22$ |
| $f : 1 \quad g : 3$ | $90.00 \pm 0.46$ | $1.860 \pm 0.12$ | $89.72 \pm 0.81$ | $1.748 \pm 0.10$ | $90.11 \pm 0.43$ | $2.595 \pm 0.38$ |

### B.6 ROBUSTNESS OF SPLITTING POINT

As discussed in the main text, the empirical coverage of Feature CP is pretty robust to the splitting point. We show both experiments of small neural networks (Table 9) and large neural networks (Table 10). We remark that one can also apply standard cross-validation to find the best splitting point, which may return a shorter band.

### B.7 FCP IN CLASSIFICATION

In this section, we show how to deploy FCP in classification problems. The basic ideas follow Algorithm 3, except for two points:

1. We use cross-entropy loss instead of MSE loss when calculating the non-conformity score.

2. We do not deploy LiPRA, but use a sampling method to return the final confidence band, which induces a looser bound.

For each sample in the calibration fold, we sample 1000 samples in the confidence band of feature space, and return the corresponding predictions in output space to get the final confidence band. We summarize the experiment results in Table 2, where we follow the statistics of APS and RAPS in Angelopoulos et al. (2021b).

### B.8 THE VALIDATION OF NON-CONFORMITY SCORE

**Distribution for non-conformity score in calibration fold.** We plot the distribution of calibration score in Figure 8. We plot each non-conformity score in the calibration fold. The distribution of non-conformity scores is smooth and single-peak in the real-world dataset, meaning that the proposed score is reasonable.

Besides, one may wonder what happens if the predicted function is close to the ground truth function, which may lead to a zero quantile of the non-conformity score. We conduct experiments to show that zero quantiles would not underestimate the final non-conformity score. Specifically, we create a dataset that the prediction target is given by $y = f^*(x) + \epsilon$, where $f^*$ denotes a three-layer neural network, $\epsilon = 0$ with probability 0.901 and $\epsilon$ follows a standard Gaussian with probability 0.009.

Table 10: Ablation study of the number of layers in $f$ and $g$ ($\alpha = 0.1$) in large neural networks on ImageNet.

| SPLITTING POINT (PREDICTION HEAD $g$) | COVERAGE | LENGTH |
|---|---|---|
| THE LAST LAYER | $0.900 \pm 0.0018$ | $3.32 \pm 0.10$ |
| LAST TWO LAYERS | $0.901 \pm 0.0047$ | $3.26 \pm 0.03$ |
| LAST THREE LAYERS | $0.900 \pm 0.0029$ | $3.30 \pm 0.01$ |

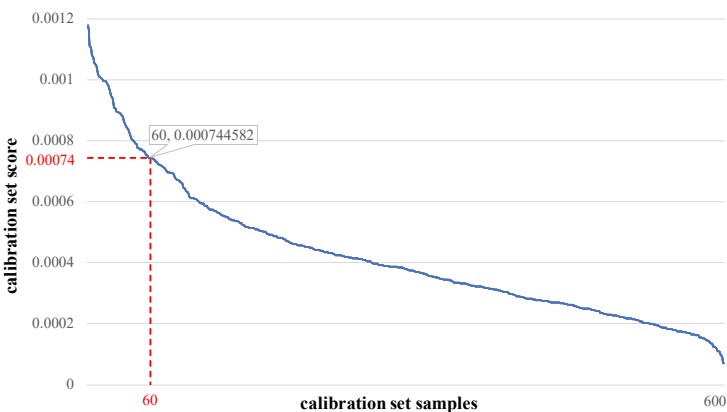

Figure 8: The distribution of the calibration score for the segmentation task. This indicates that the definition of our non-conformity score is a proper one.

Here the $0.901$ is to let the 90% quantile of the non-conformity score on a calibration set to be $0$. Besides, we make the predicted model satisfy $\hat{f} = f^*$. We summarize the results (with $\alpha = 0.1$) as follows (five repeated experiments with different $f^*$). Here $x$ is generated with a uniform distribution between $0$ and $1$. Note that the zero-length is implied by zero quantiles.

However, in practice it is impossible for the predicted model to be $f^*$, therefore exact zero-quantile is impossible. We then summarize the results when the model $\hat{f}$ is obtained via training on a training fold. The non-zero length implies that the quantile is non-zero.

## B.9    ADDITIONAL EXPERIMENT RESULTS

This section provides more experiment results omitted in the main text.

**Visualization for the segmentation problem.** We also provide more visualization results for the segmentation problem in Figure 9.

Table 11: Zero quantile does not effects the empirical coverage ($\hat{f} = f^*$).

| TRIAL | COVERAGE | LENGTH |
|-------|----------|--------|
| 1 | 90.63 | 0 |
| 2 | 90.08 | 0 |
| 3 | 89.72 | 0 |
| 4 | 89.66 | 0 |
| 5 | 90.21 | 0 |

Table 12: Zero quantile does not effects the empirical coverage ($\hat{f} \neq f^*$).

| TRIAL | COVERAGE | LENGTH |
|-------|----------|--------|
| 1 | 90.55 | 0.1274 |
| 2 | 89.72 | 0.1105 |
| 3 | 89.75 | 0.1120 |
| 4 | 89.31 | 0.1202 |
| 5 | 90.10 | 0.1245 |

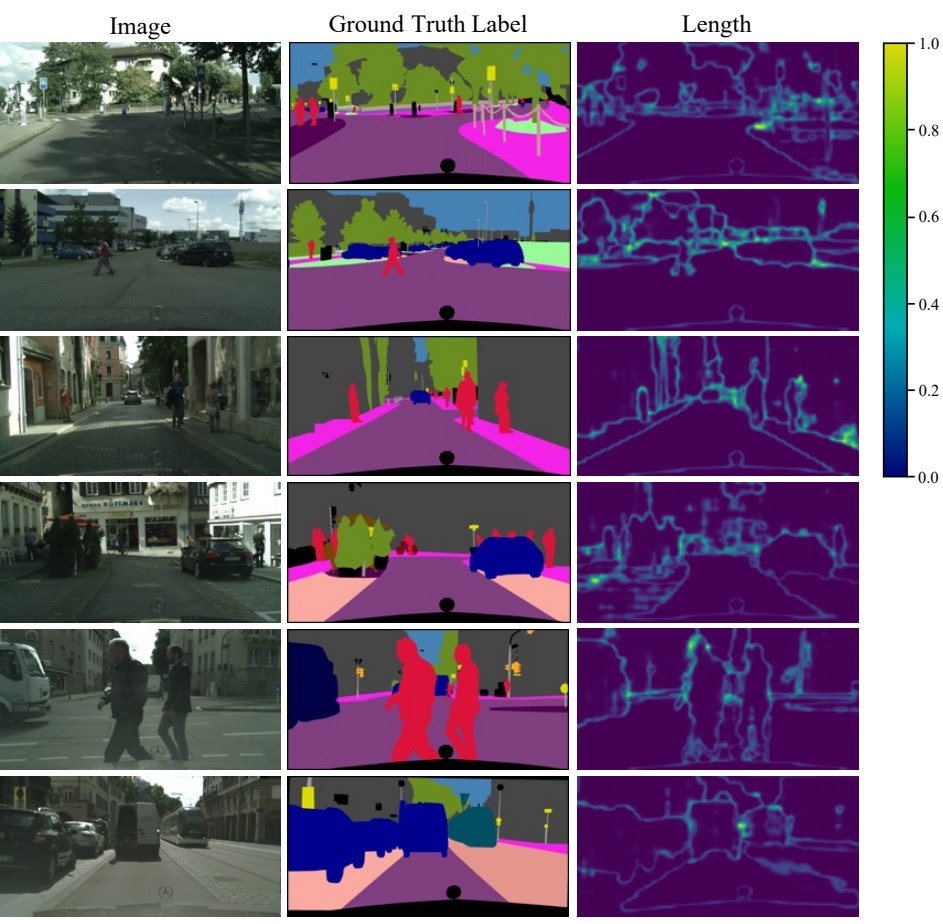

Figure 9: More visualization results for the Cityscapes segmentation task.

