# OpenReview forum: "Predictive Inference with Feature Conformal Prediction"
_ICLR.cc/2023/Conference — ICLR 2023 poster_

### Official Review · Reviewer_2teK · 2022-10-24

**Confidence:** 4
**Correctness:** 3
**Technical Novelty And Significance:** 4
**Empirical Novelty And Significance:** 2
**Recommendation:** 6

**Clarity, Quality, Novelty And Reproducibility:**

Clarity: the paper is mostly clearly written — I mentioned unclear points in the weakness discussion.

Quality: The claim is supported by theories and empirical results — I mentioned related concerns in the weakness discussion.

Novelty: I like the novel idea of applying CP in feature space; this may fertilize interesting ideas.


**Strength And Weaknesses:**

**Strengths**:
* Suggests an interesting extension for conformal prediction.

**Weaknesses**:
* Convincing arguments on the necessity of Feature CP compared to Vanilla CP is unclear.
* Incorrect statements appear; assumptions are implicit in theorem statements.


**Weakness1**: The paper claims that Feature CP is more efficient than the vanilla CP (e.g., “if we instead employ conformal prediction on the more meaningful feature space, albeit all images have the same uncertainty on this intermediate space, the pixels would exhibit effectively different uncertainty in the output space after a non-trivial non-linear transformation“). But, this justification is unclear.

1. Before achieving the efficiency of prediction sets, Feature CP needs to satisfy the coverage guarantee as in the vanilla CP. This is proven in Theorem 4. However, Theorem 4 seems to have Equation (3) solved exactly, which is not clear. If that’s true, please clearly specify in Theorem 4. If not, it would be useful to highlight this. Otherwise, this hidden assumption undermines Theorem 4.

2. I think Theorem 5 is the crux of this paper, as it explains why Feature CP can be more efficient than Vanilla CP. First, the notation $\alpha$ is reused in the theorem statement, which is very confusing as $\alpha$ is reserved for a desired coverage in CP. Please use another term if it is not intended. Also, I believe this theorem implicitly assumes a special score function, i.e., a norm of a vector. If so, please explicitly state in the theorem statement. Apart from this, I’m not sure if the cubic condition can be even satisfied in real datasets. Based on the proof, it looks like the paper made this assumption to prove the statement. If not, why is the cubic condition natural? In short, to my current understanding, Theorem 5 only holds under limited setups, thus I’m not sure if this can strongly support that Feature CP is more efficient than Vanilla CP.

3. Figure 2 and Table 1 empirically show that Feature CP is more efficient than Vanilla CP. But, the score function by Vanilla CP is too naive; for any test sample, it always returns the same length (as in Algorithm 1 “Ensure” line). One easy way to address this issue is considering the standard deviation of the prediction (e.g., Equation (1) in [R1] — this is a widely used trick to convert a point estimator to an uncertainty estimator by adding one more header for standard deviation prediction in a neural network). Can Feature CP outperform this simple baseline?

[R1] http://proceedings.mlr.press/v128/vovk20a/vovk20a.pdf

**Weakness 2**: The coverage guarantee in Equation (1) is not valid. The probability is taken over also a calibration set (e.g, Equation (1) in Tibshirani et al., 2019). The same issue appears in Theorem 4 (but fortunately, it does not affect the proof).

As mentioned in Weakness 1, each theorem statement would be better if it explicitly includes required assumptions if there is any (e.g., a requirement of a norm-style score function in Theorem 5).

In related work, the statement "Different from the above techniques, conformal prediction is appealing due to its simplicity, computationally free, and model-free properties.” is not true for calibration work. Most calibration approaches (e.g., temperature scaling) is simple, computationally free, and model-free; they simply solve a different problem.


**Summary Of The Paper:**

This paper proposes feature conformal prediction, which considers a prediction set over feature space rather than output space. In particular, the proposed novel conformal prediction is proved to satisfy the coverage guarantee under some assumptions. Moreover, the paper provides that the feature conformal prediction is provably more efficient than the vanilla CP. The efficacy (in terms of coverage guarantees and efficiency) of the feature conformal prediction is evaluated over five UCI datasets and one semantic segmentation dataset (along with one synthetic data for semantic segmentation).

**Summary Of The Review:**

I think applying CP in feature space is very interesting. But, currently I’m not fully convinced on the claim that Feature CP is more efficient than Vanilla CP, as discussed. Thus, I lean towards rejection, but hope to hear the author's opinions and additional results (if any).


=== After rebuttal

Thanks for the response. Most of my concerns are addressed, and the benefit outweighs the remaining concerns (e.g., on the justification of the cubic conditions). So, I raised my score — I like the idea of forming sets in feature space (but we can use only differentiable score functions, like DNNs, which is worth highlighting).

---

> ### Author Response · Authors · 2022-11-10
> **Response to reviews**
>
> We thank the reviewer for the helpful comments. And we are happy that the reviewer finds our proposed method novel. We have revised our manuscript accordingly based on the reviewer's opinions. Below, we do our best to address the reviewer's questions adequately such that we could receive a better score.
>
> >Feature CP needs to satisfy the coverage guarantee [...] However, Theorem 4 seems to have Equation (3) solved exactly, which is not clear.
>
> We are sorry for the confusion. We point out that Theorem 4 does not need to solve Equation 3 exactly. In fact, the steps in surrogating Equation 3 do not break exchangeability and therefore do not break Theorem 4. Therefore, there is no additional assumption hidden in Theorem 4.
>
> >Also, I believe this theorem implicitly assumes a special score function, i.e., a norm of a vector. If so, please explicitly state in the theorem statement.  [...] Apart from this, I’m not sure if the cubic condition can be even satisfied in real datasets.
>
> Thanks for the helpful advice. We have explicitly stated it in Theorem 5 in the new version (Page 6, marked in blue).
> Theorem 5 aims at providing some theoretical evidence on how feature conformal prediction exploits feature information, and we have verified the cubic condition in Table~8. We refer to the General Response for more discussion.
>
> >Figure 2 and Table 1 empirically show that Feature CP is more efficient than Vanilla CP. But, the score function by Vanilla CP is too naive; for any test sample, it always returns the same length (as in Algorithm 1 “Ensure” line). One easy way to address this issue is considering the standard deviation of the prediction (e.g., Equation (1) in [R1] — this is a widely used trick to convert a point estimator to an uncertainty estimator by adding one more header for standard deviation prediction in a neural network). Can Feature CP outperform this simple baseline?
>
> We strengthen that our feature-level methodology is a novel and orthogonal direction upon existing conformal prediction method, which generalizes given techniques. As a result, it is indeed possible to combine it with the mentioned simple baseline and see whether there is an improvement. Nonetheless, we think the current experimental comparison between CQR and Feature CQR is already enough to show the advantage of our methodology, as CQR could be considered as a more advanced adaptive CP method than the simple baseline. To put it another way, we have already shown that feature-level operation could bring benefits to both naive and complex conformal prediction methods. What's more, our method can be combined with any pretrained model as a plug-in component.  Though the mentioned simple baseline is an effective way to provide adaptive predictive interval, it would require a re-training with a slightly different model architecture as CQR does.
>
> > The coverage guarantee in Equation (1) is not valid. The probability is taken over also a calibration set (e.g, Equation (1) in Tibshirani et al., 2019). The same issue appears in Theorem 4 (but fortunately, it does not affect the proof). [...]
>
> Thanks for pointing it out! We have corrected them in the new version (Page 6, marked in blue).
>
> Once again, we appreciate the reviewer's precious time. We are eager to engage in further discussions to clear out any confusion.

---

> ### Author Response · Authors · 2022-12-07
> **Thank you!**
>
> We are happy that the reviewer likes our idea and updates the score after the rebuttal. We are eager to engage in any further discussion to help the reviewer's evaluation. Thanks!

---

### Official Review · Reviewer_9LYb · 2022-10-25

**Confidence:** 3
**Clarity, Quality, Novelty And Reproducibility:** The paper is well-written. The work i…
**Correctness:** 4
**Technical Novelty And Significance:** 3
**Empirical Novelty And Significance:** 3
**Recommendation:** 5

**Strength And Weaknesses:**

- Overall, it is a well written paper and the proposed method looks promising.
- The methodology of this paper hinges on the choice of maps $\hat g$ and $\hat f$ such that the predictor $\hat \mu$ can be represented as $\hat\mu = \hat g \circ \hat f$. While the choice of $\hat g$ and $\hat f$ may be clear for some architectures, it may be that unclear or even arbitrary for others. Take a multilayer perceptron or CNN for example, it is unclear how many last layers should be treated as $\hat g$. Different choice of $\hat g$ could lead to a different set of surrogate features, and the structure of the set of surrogate features could change the outcome of Algorithm 2. More specifically,
(a) How stable is the set of surrogate features with respect to the choice of $\hat g$? I wonder if for some choice of $\hat g$ we can’t find any surrogate features other than $\hat f$, so the only conformal score possible is zero.
(b) If the set of surrogate features can vary by different choice of $\hat g$, would this lead to different conformal scores after running Algorithm 2? I think it may be good to consider MLP of various widths, and set different number of last layers of the MLP as $\hat g$ and see if the conformal scores estimated by Algorithm 2 vary a lot.
(c) If the there is a general guideline for selecting $\hat g$?
- It looks that Algorithm 2 finds a feature $\hat u$ that minimize the difference between $Y$ and $\hat g(u)$, but I am unsure whether the $\hat u$ returned by Algorihtm 2 would be closest to $\hat f$. For example, a large learning rate may cause $u$ to be far from the initializer $\hat f$. However, a small learning rate may lead to slower convergence. Hence, in practice, how would one select the learning rate that achieve a good tradeoff between the distances of $\hat g(u)$ to $Y$ and $u$ to $\hat f$?
- I am a little bit concerned about the computational cost, because Algorithm 2 has to be done for every test output Y. The computational cost would be high if there is a large number of test points to be considered. Particularly, I guess for the Band Detection method, one would first specify a find grid for Y and run Algorithm 2 for every grid point. The computational cost should be high.


**Summary Of The Paper:**

This paper propose to construct predictive interval through confidence set on features. It is assumed that a predictor $\hat\mu$ can be written as $\hat g \circ \hat f$, where  $\hat f$ is estimated feature. This paper proposes to use conformal inference with a new conformity score to obtain a confidence set for feature $f$. Theoretical guarantees show that the new confidence interval is provably more efficient than the baseline conformal predictive interval under “cubic conditions”, which is validated empirically. Empirical experiments corroborate the merits of the proposed method.

**Summary Of The Review:**

The idea of performing conformal inference in feature space considered by this paper is a promising idea, although I do not think this paper has contributed much to the core concepts of conformal inference. For the application in deep learning, I believe there are some major questions that need to be addressed before this paper can be accepted.

---

> ### Author Response · Authors · 2022-11-10
> **Response to reviews**
>
> We thank the reviewer for the insightful comments. And we are delighted that the reviewer finds our proposed method promising. Below, we do our best to address the reviewer's questions adequately such that we could receive a better score.
>
> >the choice of $\hat{g}$ and $\hat{f}$ [...] (a) how stable is the set of surrogate features with respect to the choice of \hat{g}? I wonder if for some choice of $\hat{g}$ we can’t find any surrogate features other than $\hat{f}$, so the only conformal score possible is zero.
>
> We do the ablation study about splitting in Table 7 in the Appendix across a series of different tasks, and show that the choice of $g$ is stable.
> Besides, since the dimension of the feature is usually much larger than the dimension of the output, the surrogate features exist in most cases. In fact, note that the non-conformity score is calculated on the calibration fold. Therefore, if $\hat{f}$ is a surrogate feature, $\hat{y} = y$ in the calibration fold, and the non-conformity scores are naturally zero.
>
> >the choice of $\hat{g}$ and $\hat{f}$ [...] (b) If the set of surrogate features can vary by different choice of $\hat{g}$, would this lead to different conformal scores after running Algorithm 2?
>
> Yes, this indeed varies to different conformal scores. The explanation is that the trained function $\hat{g}$ contains the training information, and therefore the different conformal scores may relate to different training information.
>
> >the choice of $\hat{g}$ and $\hat{f}$ [...] (c) If there is a general guideline for selecting $\hat{g}$?
>
> As argued in the "General Response", we can try different potential layers and find the best one following the standard cross validation procedure.
>
> >It looks that Algorithm 2 finds a feature $\hat{u}$ that minimize the difference between $Y$ and $\hat{g}(u)$, but I am unsure whether the $\hat{u}$ returned by Algorithm 2 would be closest to $\hat{f}$. For example, a large learning rate may cause u to be far from the initializer $\hat{f}$. However, a small learning rate may lead to slower convergence. Hence, in practice, how would one select the learning rate that achieves a good tradeoff between the distances of $\hat{g}(u)$ to $Y$ and u to $\hat{f}$?
>
> Firstly, note that whether the surrogate $\hat{u}$ minimizes the difference would not influence the effectiveness of feature conformal prediction, since the exchangeability assumption always holds. From a theoretical point of view, any surrogate which does not violate the exchangeability would give a valid predictive inference result. Of course, a well optimized surrogate would intuitively bring us better empirical results (and our proposed method indeed achieve better performance in experiments). Secondly, the choice of learning rate is one of the hyperparameters that are adjusted according to the standard cross validation method. We agree that a too large/small learning rate would be improper.
>
> >I am a little bit concerned about the computational cost, because Algorithm 2 has to be done for every test output Y. The computational cost would be high if there is a large number of test points to be considered. Particularly, I guess for the Band Detection method, one would first specify a find grid for Y and run Algorithm 2 for every grid point. The computational cost should be high.
>
> A grid operation is necessary only if we want to use Band Detection to obtain a confidence band. However in practice, we use Band Estimation to obtain the band in this paper. For the setup, we only want empirical coverage (which only needs to determine whether a given Y is in the band). In that case, the computational cost for Band Detection is reasonably small as it does not require a grid operation. No matter in Band Estimation or Band Detection, we only need one forward pass of the feature extractor neural network. which is the main overhead.
>
> Once again, we appreciate the reviewer's precious time. We are eager to engage in further discussions to clear out any confusion.

---

> > ### Comment · Area_Chair_fmW9 · 2022-12-12
> > **Follow up on the response for first question**
> >
> > If the conformity score can be zero as mentioned in the response to the first question from reviewer, then one can underestimate the quantile Q_{1-alpha}. Wouldn't that be a problem?
> >
> > Please quickly respond to this question.

---

> > > ### Author Response · Authors · 2022-12-13
> > > **Responses**
> > >
> > > Dear Area Chair,
> > >
> > > Thanks a lot for the insightful question! The phenomenon that a non-conformity score can be zero also appears in the conformal prediction community and would NOT cause an underestimation of the quantile. We will make sure to add the discussion in the revision.
> > >
> > > The reason is that: if the *only* possible surrogate feature is $\hat{f}$ for a calibration sample, the argument $\hat{y} = y $ holds in the calibration fold (see Discussion A below). Therefore, since the calibration fold acts like the test fold, a zero quantile naturally leads to the claim that $1-\alpha$ testing points are well-predicted (\hat{y} = y). In such a case, confidence bands with zero length are still valid and do not cause an underestimation (see Discussion B below). Various experiment results also validate the success of surrogate features (see Discussion C below).
> > >
> > > We next provide more detailed discussions in three aspects.
> > >
> > > **Discussion A**: in FCP, the predicted feature $\hat{v}$ and the surrogate feature $v$ satisfies $\hat{g} (\hat{v}) = \hat{y}$ and $\hat{g} (v) = y$. Note that the non-conformity score measures the distance between $v$ and $\hat{v}$, and therefore a zero non-conformity score leads to $\hat{v} = v$, further leading to $\hat{y} = y$ in the calibration fold.
> > >
> > > **Discussion B**: we remark that the score is calculated based on the *calibration fold* instead of the training fold. Informally, given the training fold, the calibration fold acts similarly to the test fold due to the exchangeability assumption. Therefore, a zero quantile of score in calibration fold directly means that $1-\alpha$ samples are naturally well-predicted by the base model. In this case, confidence bands with zero length already guarantee $1-\alpha$ coverage, which is valid. Such discussion also holds for vanilla conformal prediction.
> > >
> > > **Discussion C**: Various experiment results show the effectiveness of the surrogate feature since the bands returned by FCP generally enjoy shorter lengths than vanilla CP. Generally, a surrogate feature can be found successfully in experiments because the feature is usually high-dimensional.
> > >
> > > ***Overall, both empirical and theoretical evidence shows the effectiveness of the surrogate feature.*** It is also a promising direction to find more types of surrogate features that further improve the efficiency and group coverage in conformal prediction.
> > >
> > > Best!

---

> ### Author Response · Authors · 2022-12-07
> **Do our responses clarify your concerns?**
>
> Dear reviewer,
>
> We would like to thank the reviewer again for the precious time and constructive comments. Besides, we would like to express our great gratitude for the reviewer's positive comments on the paper's novelty.
>
> Unfortunately, we have not received any feedback up to now. May we ask whether our responses have successfully clarified the reviewer's concerns? If so, we would sincerely appreciate an increase in the score.
>
> We are eager to engage in any further discussions to help the reviewer's evaluation. Thanks!
>
> Best!

---

> ### Author Response · Authors · 2022-12-12
> **Do our responses clarify your concerns?**
>
> Dear Reviewer 9LYb,
>
> We hope it does not disturb you. Once again, we appreciate your thoughtful and enlightening remarks. We would appreciate additional feedback from you, as there is only 1 day remaining before the rebuttal phase concludes. We believe our responses have allayed your concerns. Based on your and the other reviewers' suggestions, we think the manuscript now meets higher standards. If necessary, we would be happy to have any additional discussions.
>
> Best regards, Authors

---

### Official Review · Reviewer_5bhg · 2022-10-26

**Confidence:** 4
**Correctness:** 3
**Technical Novelty And Significance:** 3
**Empirical Novelty And Significance:** 3
**Recommendation:** 6

**Clarity, Quality, Novelty And Reproducibility:**

The paper is clearly written in terms of the methodology, however, the theory part is hard to digest and there are no intuitions on what the conditions are for theorem 5 to hold. In terms of reproducibility, I would like to hear the author's responses to my above questions in case I missed something in the main/appendix.
The paper is also novel in the sense that they consider different score functions for conformal prediction

**Strength And Weaknesses:**

I will split my review into strengths as well as weaknesses in this review.

I will start off with the strengths:
- The paper proposed a new way to thinking about conformal prediction by looking at perturbation in the feature space to construct a score function.
- This is a new way to construct a score function to the best of my knowledge and interesting given their experimental results.
- Given that they work in the feature space the problem hence lies in how to convert these to the output space and they propose two methods to do so.
- Experimental results look really good especially Table 1 ~40 -> ~1 seems like a huge improvement while keeping the coverage.


The weakness of the papers are the following:
- Firstly the explanations of the cubic conditions for theorem 5 are not very intuitively explained and I am left to just guess whether these conditions ever hold in practice. Could you please clarify how you determine if any of these conditions ever hold in practice? The reason I am asking this is that proofs can become trivial if one gives themselves too strong conditions and hence I will rely on other reviewers' expertise on this theoretical part of the paper.
- Next, I am confused to how to use your algorithm in the classification setting. Looking at eq 3 wouldn't there be many $v$ for which we have the discrete label Y? In that case how do we choose which $v$ and therefore which score? I might have misunderstood sth, so please clarify this part please. My concern is that given that there is an optimization step, in alg 2 how do you determine how many steps to go and the step size? These seem to be user choices which I hope the authors can point me towards in the appendix or in this rebuttal.
- The authors never discuss or even give intuition into how " (LiPRA) (Xu et al., 2020)" works and hence should be added. Please let me know if I missed it in the appendix or main script
-EXPERIMENTS: Could the authors elaborate on Table 1 and how they achieved such amazing results on length 40 -> 1 . Especially I am curious why the l_inf norm was used throughout the paper and not standard L2. The authors mention this "Secondly, an intuitive explanation relates to our usage of l∞ to form the non-conformity score during the training." I don't understand this part, and hence a clarification on the loss functions or pointing me to the appendix would be helpful in understanding the results.
- EXPERIMENTS: Fig 4 seems pointless without the interval lengths? Could you please point me to the interval lengths with varying alpha in case I missed it?
- EXPERIMENTS: Table 1 literally contradicts your assumption that your lengths should be smaller. The std are overlapping in the table and hence I wonder if the method doesn't even work on synthetic data how can it possibly work on real data?
- EXPERIMENTS: Lastly, my concern also lie in the way the authors chose the layers on which the compute the scores. The authors have some experiments in the appendix showing the picking different layers does impact the performance of their method, however, I don't see a comparison to standard CP. Does that mean if you choose the wrong layer standard CP is better? and if yes can you tell me which part of theorem 5 would be violated in that case? Could the authors please add experiments where the layer choice has been well justified and a more thorough ablation study has been done on how to choose the layer in the first place?


Overall, i quite like the idea and would happily increase me score if the above have been answered.


**Summary Of The Paper:**

This paper looks at a different way to perform conformal prediction by proposing a new score function as well as guarantees that their proposed method achieves smaller interval lengths. They prove that under some conditions (not very well explained) their proposed method can achieve smaller interval lengths compared to standard CP.

They validate their method on various datasets


**Summary Of The Review:**

I have clearly outlined my thoughts as well as concerns in the above review.
I am more than happy to increase my score in case the authors are able to clarify the misunderstanding as well questions I have outlined above.

---

> ### Author Response · Authors · 2022-11-10
> **Response to reviews**
>
> We thank the reviewer for the detailed and insightful comments. Below, we do our best to address the reviewer's questions adequately such that we could receive a better score.
>
> >Firstly the explanations of the cubic conditions for theorem 5 are not very intuitively explained and I am left to just guess whether these conditions ever hold in practice. [...]
>
> We thank the reviewer for the kind advice. We indeed have verified the cubic conditions in practice in Table 3. We also refer to the "General Response" for more discussion.
>
> >Next, I am confused to how to use your algorithm in the classification setting. [...]
>
> Indeed, the proposed algorithm cannot be directly applied to classification settings in its current shape, due to the restrictions of vanilla regression CP.  Instead, we can simply replace the base model from vanilla CP with any of its classification variants (e.g., [1, 2]), then the feature-level techniques can be deployed on the classification tasks. We leave the extension for future work.
>
> [1] Uncertainty Sets for Image Classifiers using Conformal Prediction. Anastasios Angelopoulos, Stephen Bates, Jitendra Malik, and Michael I. Jordan.
>
> [2] Classification with Valid and Adaptive Coverage. Yaniv Romano, Matteo Sesia, and Emmanuel J. Candès.
>
> >The authors never discuss or even give intuition into how " (LiPRA) (Xu et al., 2020)" works and hence should be added. [...]
>
> LiPRA transforms the certification problem as a linear programming problem, and solves it accordingly. We will add more description about it in the final version. We use the LiPRA technique to transform an interval on feature space to output space. The interval LiPRA outputs in the output space is larger than the exact results, thus our method is still theoretically valid. From an experimental view, it is sufficient for feature CP to outperform vanilla CP. However, it is not a unique method and could be replaced.
>
> >EXPERIMENTS: Could the authors elaborate on Table 1 and how they achieved such amazing results on length 40 -> 1 .  Especially I am curious why the l_inf norm was used throughout the paper and not standard L2.
>
> As stated in the last but one paragraph in Page 9 (Discussion), the reason for 40->1 is that: vanilla conformal prediction performs pretty badly on the high-dimensional problem. The reason is that in vanilla conformal prediction (1) each figure has the same band length for the vanilla version, and (2) each pixel has the same band length under \ell_inf norm. Fortunately, Feature CP can avoid these issues. Figure 3 shows that Feature CP extracts some interesting information in image segmentation problems.
>
> Besides, we use \ell_inf norm instead of \ell_2 since \ell_inf norm can easily decompose between each pixel, which is convenient for (1) the visualization (Figure 3), and (2) proposing weighted band length (Appendix B.1).
>
> >EXPERIMENTS: Table 1 literally contradicts your assumption that your lengths should be smaller. The std are overlapping in the table and hence I wonder if the method doesn't even work on synthetic data how can it possibly work on real data?
>
> Sorry for the confusion. The box-plot here does not contain the std information. We refer to the Appendix (Table 5 and Table 6) for the concrete numbers. It turns out that in most cases the bands would not overlap.
>
> >EXPERIMENTS: Lastly, my concern also lie in the way the authors chose the layers on which the compute the scores. The authors have some experiments in the appendix showing the picking different layers does impact the performance of their method, however, I don't see a comparison to standard CP. Does that mean if you choose the wrong layer standard CP is better? and if yes can you tell me which part of theorem 5 would be violated in that case? Could the authors please add experiments where the layer choice has been well justified and a more thorough ablation study has been done on how to choose the layer in the first place?
>
> For the splitting criterion, we refer to "General Response" for a discussion. Indeed, one may even get a worse band length if choosing the wrong layer, but hopefully, one can check all the potential layers and choose the best one. If the results are not satisfying, with a high probability, Statement 2 in the cubic condition is violated, since it is at the core of the cubic condition. We added an experiment to show that Statement 2 is indeed closely related to the returned band length (see General Response).
>
> Once again, we appreciate the reviewer's precious time. We are eager to engage in further discussions to clear out any confusion.

---

> > ### Comment · Reviewer_5bhg · 2022-12-11
> > **Response**
> >
> > First of all, sorry for the late reply as I was away for a personal matter and then got sick and had to stay in bed until now ...
> >
> > I have read the rebuttal and my only concern remains on the splitting. If the authors could add this "validation step" in the final version of their paper (with ALL their networks + code), together with the clarification given above, I suggest accepting this paper.
> >
> > Lastly, I would also like the authors to add the L2 norm in the appendix for illustration purposes.
> >
> > Best

---

> > > ### Author Response · Authors · 2022-12-11
> > > **Responses**
> > >
> > > Dear Reviewer,
> > >
> > > We apologize for any inconvenience. We hope that you recover soon. We will ensure that the required information is included in the final version. We are truly grateful for the valuable time and constructive feedback provided by the reviewer.
> > >
> > > Best!

---

> ### Author Response · Authors · 2022-12-07
> **Do our responses clarify your concerns?**
>
> Dear reviewer,
>
> We would like to thank the reviewer again for the precious time and constructive comments. Besides, we would like to express our great gratitude for the reviewer's positive comments on the paper's novelty.
>
> Unfortunately, we have not received any feedback up to now. May we ask whether our responses have successfully clarified the reviewer's concerns? If so, we would sincerely appreciate an increase in the score.
>
> We are eager to engage in any further discussions to help the reviewer's evaluation. Thanks!
>
> Best!

---

### Official Review · Reviewer_MB3r · 2022-10-27

**Confidence:** 3
**Correctness:** 4
**Technical Novelty And Significance:** 2
**Empirical Novelty And Significance:** 3
**Recommendation:** 6

**Clarity, Quality, Novelty And Reproducibility:**

The paper is written in a clear way, with both methods, theory, and numerical studies supported.

One point with respect to the clarity is that the authors should clarify which norm they are using in the paper, for example, Section 4.1 etc.

**Strength And Weaknesses:**

Strength:
The authors aim to improve the original conformal prediction (CP) method by leveraging the idea of semantic feature spaces. The idea is novel and useful. They provided a solid theory on both the effective aspect (coverage) and efficiency (confidence band length). In addition, they compared their method with vanilla CP in various scenarios, validating their claims. They also provided many examples to demonstrate their theoretical statements.

Weaknesses:

1. The idea of conducting CP in semantic feature spaces is really interesting. However, the authors had little discussion on how we split f and g. The two functions f and g should be indistinguishable. I have this concern because it seems very important to choose an appropriate feature space.

2. There are several "surrogate" steps throughout the method. They used $\hat{u}$ to replace the ground truth label, utilized the gradient descent method to calculate the non-conformity score, and employed "Band Estimation" and "Band Detection" to approximate the confidence band in output space. As for the theoretical guarantees, do the authors need the assumption of the surrogate estimators?

3. Numerically, the authors could consider comparing their method with more baselines. There are a lot of improvements to vanilla CP in the literature. Beating only the original one may not be convincing enough.


**Summary Of The Paper:**

The authors propose feature conformal prediction for semantic feature spaces by leveraging the inductive bias of deep representation learning.


**Summary Of The Review:**

Overall, the paper is well written with both theoretical and numerical guarantee.

---

> ### Author Response · Authors · 2022-11-10
> **Response to reviews**
>
> We thank the reviewer for the insightful and supportive comments. Below, we do our best to address the reviewer's questions adequately such that we could receive a better score.
>
> >The idea of conducting CP in semantic feature spaces is really interesting. However, the authors had little discussion on how we split f and g. The two functions f and g should be indistinguishable. I have this concern because it seems very important to choose an appropriate feature space.
>
> Thanks for the insightful question, and we refer to "General Response" for the discussion.
>
> >There are several "surrogate" steps throughout the method. They used $\hat{u}$ to replace the ground truth label, utilized the gradient descent method to calculate the non-conformity score, and employed "Band Estimation" and "Band Detection" to approximate the confidence band in output space. As for the theoretical guarantees, do the authors need the assumption of the surrogate estimators?
>
> Thanks for pointing it out! The effects of these surrogates on the theorem can be split into two folds. For the first theorem (which shows the effectiveness), all these surrogates are valid because they maintain the exchangeability assumption. For the second theorem (which shows the efficiency), we implicitly restrict these surrogates by cubic condition (argument 1 shows that Feature CP does not cost much individually, which implicitly bound the cost of these surrogates).
>
> >Numerically, the authors could consider comparing their method with more baselines. There are a lot of improvements to vanilla CP in the literature. Beating only the original one may not be convincing enough.
>
> Thanks for the kind advice. The feature-level technique is general and could be adopted onto any existing conformal prediction framework, thus it is an orthogonal direction with regards to existing baselines. We show that they can be deployed into the variants of vanilla CP. For example, we compare CQR with Feature CQR (Figure 2 and Table 5). We agree with the argument that investigating the combination between Feature CP and the existing method will be a promising future direction.
>
> >One point with respect to the clarity is that the authors should clarify which norm they are using in the paper, for example, Section 4.1 etc.
>
> Here we use the infinity norm in the experiment part. But all other norms are also valid in the algorithm.
>
> Once again, we appreciate the reviewer's precious time. We are eager to engage in further discussions to clear out any confusion.

---

### Author Response · Authors · 2022-11-10
**General Response**

We appreciate the reviewers for their constructive comments that help to enhance our manuscript considerably. Furthermore, we would like to express our gratitude to the reviewers for their positive feedback on the paper's novelty, which has been a great source of inspiration for us. The critical comments are mainly around two points (1) the splitting criterion of feature f and prediction head g, and (2) the intuition on cubic conditions. Below, we first focus on the comments raised by most of the reviewers and then address each reviewer individually.

### On the splitting criterion of feature $f$ and prediction head $g$:
As mentioned in the main text (the last paragraph of page 9), the splitting criterion is non-trivial. However, experiment results of Feature CP on different splitting points stably outperform vanilla CP. ***Fortunately, the splitting is a hyperparameter, and we can use standard cross validation to choose its value.*** Since the output space is one of the special feature layers, such techniques always generalize the scope of vanilla CP.

### On the intuition behind cubic conditions:
1.  As mentioned in the introduction, at a colloquial level, the cubic conditions assume the feature space has a smaller distance between individual non-conformity scores and their quantiles, which reduces the cost of the quantile operation. This is the key observation of cubic condition (argument 2).
2. We conduct an experiment to show the above argument in Table 3 in Appendix. We fix some flexible parameters to ease the experiment (e.g., we relax the Holder Continuous Assumption to Lipschitz continuous). Table 3 indeed shows that feature space has a smaller distance between individual non-conformity scores and their quantiles.
3. In the rebuttal period, we also conduct experiments (Page 25, Figure 9) to show that cubic condition is indeed closely related to the performance of Feature CP. For each given dataset, the cubic metric is negatively related to the band length, and thus positively related to the efficiency.

**Cubic metric is negatively related to the band length**

|            |          | length  | cubic metric  |
|  :----:  | :----:  | :----:  | :----:  |
| bio  | f:1 g:3 | 1.86 | 1.00 |
|        | f:2 g:2 | 1.87 | 0.81 |
|        | f:3 g:1 | 1.96 | 0.75 |
| bike  | f:1 g:3 | 1.75 | 0.75 |
|            | f:2 g:2 | 1.79 | 0.70 |
|            | f:3 g:1 | 1.92 | 0.69 |
| blog  | f:1 g:3 | 2.60 | 2.72 |
|          | f:2 g:2 | 2.81 | 2.37 |
|          | f:3 g:1 | 3.32 | 1.81 |



### On the contribution
The contribution of this paper is to propose feature-level conformal inference technique, which is (a) efficient, (b) simple but general, and (c) contains semantic training information. For (a), we show that Feature CP achieves smaller band lengths uniformly over many real-world datasets (Fig. 2). For (b), we show that such feature-level techniques can be generalized to many variants of vanilla CP, e.g., CQR (Appendix B.4). For (c), we conduct feature CP on both well-trained and untrained networks and observe that Feature CP performs well only on well-trained networks in Table 4.

---

### Comment · Area_Chair_fmW9 · 2022-12-11
**Relation to localized conformal prediction and prediction sets for classification tasks**

Dear Authors,

Couple of comments and questions. I posted them last week, but unfortunately didn't realize authors couldn't see.

1. Can you please discuss the relation and/or implication of feature conformal prediction with the framework of localized conformal prediction?

Original paper: https://arxiv.org/abs/2106.08460 NeurIPS-2021 paper that demonstrates improved efficiency for real-world regression tasks: https://arxiv.org/abs/2106.00225

2. Can you discuss the generality of the approach for classification tasks to obtain prediction sets?

Can you also share any experimental results (e.g., on CIFAR100) comparing with prior methods such as APS and RAPS? See https://arxiv.org/abs/2009.14193 for code.

If you get a chance, please respond to them. I understand if you cannot given the limited time.

Thanks!

---

> ### Author Response · Authors · 2022-12-12
> **Response to AC**
>
> Dear Area Chair,
>
> Thanks a lot for the questions! Due to the time limit (we only have one day till the rebuttal period ends), we are not confident that we can complete all the required experiments. We here quickly provide some discussion and intuition instead, and will post the experiment details as soon as possible. We promise to add all the discussions and the ongoing experiment results in the final revision.
>
> Regarding Problem 1, Localized Conformal Prediction (LCP) and Feature Conformal Prediction (FCP) can both return confidence bands with different lengths. However, they are different in the following two points:
>
> - Different goals. The motivation of LCP is to improve group coverage, by weighing more on the calibration samples around the testing point. Differently, the motivation of FCP is to improve efficiency (band length) using feature information. One can simultaneously use the feature-level and localization-level techniques to improve both group coverage and band length (see details below).
>
> - Different tools. LCP mainly uses a *weight* to adjust the non-conformity score. Differently, FCP uses a *predicting head* for adjustment. In LCP, the weight can be either untrained [1] or trained [2], while in FCP, the predicting head should be trained to gain more information (see Table 4 in Appendix).
>
> ***More importantly, the two techniques are orthogonal, and one can combine the two techniques to improve both group coverage and band length.*** For example, one can introduce weight (just like [1, 2]) in the quantile step of FCP (Step 4 in Algorithm 3, main text in FCP). One can calculate the weight in either the input or the feature space. We believe such localization operation has the potential of further improving the group coverage of FCP.  We will make sure to add the discussion and the experiment details in the revision.
>
> Regarding Problem 2, feature-level operations can be naturally applied in the classification regimes as follows. To generalize FCP to classification tasks, one need to redefine *surrogate features* (see Step 3 in Algorithm 3, main text). As the simplest case, the surrogate feature $\hat v$ could be optimized such that the classification cross entropy loss (rather than the regression loss) is small enough, i.e., $CrossEntropy(\hat{g}(\hat{v}), y) \leq \epsilon$. Here $\epsilon$ is a small threshold to make the prediction based on $\hat v$ be close to the label $y$ (we do not need this to be exact, since the predictive interval would hold valid as long as the procedure does not violate the exchangeability as in Theorem 4). Other steps of Feature CP still hold accordingly. As you have mentioned, one can further use the ideas in [3] to define more complex surrogate features to improve efficiency, e.g., features such that the largest two labels contain the ground truth label. We will add the experiments on classification benchmarks in the revision.
>
> Best,
> Authors
>
> [1] Localized Conformal Prediction: A Generalized Inference Framework for Conformal Prediction, Leying Guan
> [2] Locally Valid and Discriminative Prediction Intervals for Deep Learning Models, Zhen Lin, Shubhendu Trivedi, Jimeng Sun
> [3] Uncertainty Sets for Image Classifiers using Conformal Prediction, Anastasios Angelopoulos, Stephen Bates, Jitendra Malik, Michael I. Jordan

---

### Decision · Program_Chairs · 2023-01-20

**Decision:**

Accept: poster

**Justification For Why Not Higher Score:**

There are some outstanding issues as mentioned in my meta-review. So I don't think the paper deserves a higher score.



**Justification For Why Not Lower Score:**

Novelty of the idea behind feature CP and theoretical analysis are good. This is why we agreed to give an accept.

**Metareview: Summary, Strengths And Weaknesses:**

This paper considers the problem of uncertainty quantification (UQ) of deep neural networks for regression tasks within the conformal prediction (CP) framework. It takes a different approach for CP by proposing feature conformal prediction, which considers a prediction set over feature space rather than output space. Theoretical guarantees show that the prediction interval is more efficient than the baseline CP under some conditions. Experimental results demonstrate the merits of the proposed feature CP approach.

Overall, the reviewers' liked the feature CP approach, but also raised some questions about design choices and ablation experiments. The authors' response addressed most of the concerns except a few which were left partly because reviewers' could not respond on time due to health issues.

Some outstanding issues that need to be addressed in the final paper if it gets accepted.
1. There are other improvements to vanilla CP including localized CP. The paper should consider adding stronger baselines, discussion about localized CP, and potentially show that feature CP and localized CP can be synergistically combined as mentioned in the response.
2. The discussion about choice of maps \hat{g} and \hat{f} is an important one for both theory and practice. So I strongly encourage the authors' to add this discussion and guidance for practitioners for ease-of-use. Also, clarify this design choice for different types of deep neural networks and potentially consider adding ablation results in the main paper. Please add the discussion about the impact of conformity score being zero on the estimation of quantile Q_{1-alpha} and also provide empirical evidence to backup those claims (as mentioned in the response).
3. Please address the concern about "splitting" from one reviewer and results on all networks (small and big) to demonstrate robustness.
4. There should be some discussion about the extension of feature CP for classification tasks and good results on classification benchmarks. If this is considered for future work and no experiments are added to the paper, I recommend restricting the scope of the problem setup to regression tasks.

Authors' were give some more extra time and they have addressed #2, #3, and #4 reasonably. So I recommend accepting the paper. I strongly encourage the authors to address the above issues and reflect the author-reviewer discussion in the final paper.

**Note From Pc:**

if the above contains the word "oral" or "spotlight" please see: "oral" presentation means -> notable-top-5% and "spotlight" means -> notable-top-25%. As stated in our emails, we are disassociating presentation type from AC recommendations

**Summary Of Ac-Reviewer Meeting:**

I sent emails to all reviewers for discussion and virtual meeting 2 weeks back -- one reviewer never responded until now, another reviewer got sick after NeurIPS trip and responded only yesterday. So in the end, I was only able to have email discussion about outstanding issues. I got input from three out of four reviewers. I also posted couple of questions based on my own reading of the paper and followed up on outstanding questions.

The main issues raised were:
- Design choices in feature CP about the maps \hat{f} and \hat{g}. Authors provided discussion and suggested this can be done similar to tuning hyper-parameters in ML and provided ablation results in the Appendix.
- Clarification on cubic conditions for theoretical analysis of efficiency. Authors responded with more intuition and empirical results to support.
- Clarification on splitting and experiments on all networks. Authors promised to address in the final paper and reviewer was okay with accepting the paper.
- Relation to localized CP. Authors responded by saying that their method is orthogonal and can be combined with LCP. This need to be addressed in the final paper.
- Extension of the approach for classification tasks. Authors suggested this is left for future work.